# Distinct elongation stalls during translation are linked with distinct pathways for mRNA degradation

**Anthony J Veltri**[1†], **Karole N D'Orazio**[1‡], **Laura N Lessen**[1§], **Raphael Loll-Krippleber**[2], **Grant W Brown**[2], **Rachel Green**[1]*

[1]Department of Molecular Biology and Genetics, Howard Hughes Medical Institute, Johns Hopkins University School of Medicine, Baltimore, United States; [2]Department of Biochemistry and Donnelly Centre, University of Toronto, Toronto, Canada

**\*For correspondence:**
ragreen@jhmi.edu

**Present address:** [†]Versiti Blood Research Institute, Milwaukee, United States; [‡]Regeneron Pharmaceuticals, New York, United States; [§]GlaxoSmithKline, Rockville, United States

**Abstract** Key protein adapters couple translation to mRNA decay on specific classes of problematic mRNAs in eukaryotes. Slow decoding on non-optimal codons leads to codon-optimality-mediated decay (COMD) and prolonged arrest at stall sites leads to no-go decay (NGD). The identities of the decay factors underlying these processes and the mechanisms by which they respond to translational distress remain open areas of investigation. We use carefully designed reporter mRNAs to perform genetic screens and functional assays in *Saccharomyces cerevisiae*. We characterize the roles of Hel2, Syh1, and Smy2 in coordinating translational repression and mRNA decay on NGD reporter mRNAs, finding that Syh1 and, to a lesser extent its paralog Smy2, act in a distinct pathway from Hel2. This Syh1/Smy2-mediated pathway acts as a redundant, compensatory pathway to elicit NGD when Hel2-dependent NGD is impaired. Importantly, we observe that these NGD factors are not involved in the degradation of mRNAs enriched in non-optimal codons. Further, we establish that a key factor previously implicated in COMD, Not5, contributes modestly to the degradation of an NGD-targeted mRNA. Finally, we use ribosome profiling to reveal distinct ribosomal states associated with each reporter mRNA that readily rationalize the contributions of NGD and COMD factors to degradation of these reporters. Taken together, these results provide new insight into the role of Syh1 and Smy2 in NGD and into the ribosomal states that correlate with the activation of distinct pathways targeting mRNAs for degradation in yeast.

## Editor's evaluation

This study provides a broad comparison of the roles of protein factors in No-Go Decay (NGD) and Codon-Optimality-Mediated Decay (COMD) in the yeast, *S. cerevisiae*. A major strength of the manuscript is the direct comparison between one mRNA with a single strong translational stall and another similar mRNA with many slow translation sites (caused by changes in the genetic code). The analysis of both the factors that cause decay of these mRNAs as well as the ribosome states on the different mRNAs increases our understanding of the molecular basis for the different mechanisms of mRNA quality control. The results support a model that these are largely distinct processes driven by different protein factors in response to different ribosome conformations and, in addition, the work identifies several differences between yeast and human NGD and COMD.

## Introduction

Translation of mRNAs to produce proteins is a fundamental cellular process that supports the cell's ability to carry out the basic enzymatic reactions needed for life. To prevent errors that arise during

this complex process from compromising cellular metabolism, specialized molecular pathways have evolved to recognize and regulate problematic translation events (*D'Orazio and Green, 2021a*; *Inada, 2017*; *Yan and Zaher, 2019*). These mechanisms are coupled to RNA decay pathways that target problematic mRNAs and prevent continuing diversion of ribosomes toward unproductive translation. The set of factors involved in this crucial recognition of problems that arise during translation elongation and the mechanisms by which they exert their downstream effects on mRNA stability remain only partially characterized.

General mRNA decay in yeast is catalyzed primarily by mRNA decapping and 5' to 3' exonucleolytic degradation by Xrn1, while the 3' to 5' exonuclease (the exosome) is thought to play a role only under certain circumstances (*Muhlrad et al., 1994*). Recent foundational work in yeast, and subsequently in zebrafish and mammals, discovered that mRNA stability is correlated with its codon usage (*Presnyak et al., 2015*; *Mishima and Tomari, 2016*; *Wu et al., 2019a*): mRNAs enriched in non-optimal codons have short half-lives and are rapidly degraded by the cytoplasmic Ccr4-Not deadenylation complex and the decapping activator Dhh1 (*Radhakrishnan et al., 2016*; *Webster et al., 2018*; *Sweet et al., 2012*). Recent biochemical and structural evidence supports a model in which suboptimal codons in the ribosomal A site slow down translation elongation, allowing deacylated tRNA to diffuse away from the E site and enabling the critical adaptor protein Not5 to bind. Not5 binding in the vacant ribosomal E sites can recruit the Ccr4-Not complex to promote mRNA deadenylation, decapping, and decay (*Buschauer et al., 2020*). These observations provide molecular insight into codon-optimality-mediated decay (COMD) and support for the idea that this pathway represents an important determinant of general cellular mRNA half-lives.

In contrast to the normal slowing of translation that occurs transiently as ribosomes decode less optimal codons, the cell also possesses quality-control machinery to resolve more deleterious ribosomal stalls that can arise from chemical damage in the mRNA (truncation, depurination, nucleobase dimers, oxidative damage, etc.), difficult to unwind secondary structure, or incorrect nuclear mRNA processing events (*D'Orazio and Green, 2021a*). Adjacent pairs of specific rare codons can mimic these events and induce strong inhibition of translation elongation and associated mRNA decay in *Saccharomyces cerevisiae* (*Gamble et al., 2016*). For example, consecutive CGA codons induce terminal stalls and have been routinely included in reporter mRNAs to trigger an mRNA surveillance pathway referred to as no-go decay (NGD) (*Tsuboi et al., 2012*; *Letzring et al., 2013*; *Tesina et al., 2020*). On these problematic mRNAs, ribosomes stall on the CGA codons, leading to ribosomal collisions that promote small-subunit protein ubiquitination by the E3 ligase Hel2 (mammalian ZNF598) (*Juszkiewicz et al., 2018*, *Matsuo et al., 2017*, *Saito et al., 2015*, *Sundaramoorthy et al., 2017*), ribosomal clearance by the helicase Slh1 and ribosome quality control trigger (RQT) complex (*Ikeuchi et al., 2018*), and nascent peptide decay by the ribosome quality control (RQC) complex (*Brandman et al., 2012*). The accumulation of colliding ribosomes is thought to trigger decapping and Xrn1-mediated mRNA degradation, although the specific molecular players and interactions responsible for triggering this decay remain poorly defined (*D'Orazio et al., 2019*; *Simms et al., 2018*). Additionally, under conditions where the ribosome rescue machinery is compromised or overwhelmed, cleavage of reporter mRNAs by the endonuclease Cue2 and Dom34-mediated rescue provides an alternate route for mRNA degradation and ribosome rescue (*Doma and Parker, 2006*; *D'Orazio et al., 2019*; *Glover et al., 2020*). NGD, as a 'quality control' pathway, is thought to minimize the detrimental effects from damaged or problematic mRNAs in the cell and to reduce the overall impact of proteotoxic stress (*Brandman et al., 2012*; *Ishimura et al., 2014*; *Martin et al., 2020*).

Although NGD and COMD can be triggered by seemingly similar mRNA sequences to converge on Xrn1-mediated exonucleolytic decay of mRNAs (*Pelechano et al., 2015*), the extent to which these processes overlap in specificity and activity remains unclear, as do their complete sets of accessory factors. Moreover, the molecular states of the ribosome which define and activate these pathways have not been systematically compared. In this study, we address these questions through genetic screening and functional assays in the yeast *S. cerevisiae*. We use reporter mRNAs designed to trigger NGD or COMD and perform reporter-synthetic genetic array (R-SGA) screens to identify factors critical to these separate mRNA decay pathways. Importantly, we identify a critical role for Syh1 as a strong effector for decay of mRNAs with terminal stalls. We use flow cytometry and northern blotting combined with genetic perturbations to reveal the contributions of other NGD factors and COMD factors to translational repression and decay of the reporter mRNAs. Finally, we use ribosome profiling

of NGD and COMD reporters to isolate the activities of the major players in these pathways and connect these activities to the molecular states of elongating ribosomes. These data provide a basis for understanding the unique contributions of each of these pathways to translation-coupled mRNA decay and contextualizes their effects in the larger cellular process of translation surveillance.

## Results

### A genetic screen identifies NGD factors in yeast

To identify protein factors that contribute to NGD, we developed reporter constructs with well-defined sequence features designed to trigger ribosome stalling and associated quality control. Reporter mRNAs were under the control of the inducible, bidirectional *GAL1-10* promoter and encoded GFP followed by either a fully codon-optimized yeast *HIS3* gene (termed GFP-OPT) or *HIS3* interrupted by 12 repeats of the highly non-optimal CGA codon (termed GFP-CGA; *Figure 1A*). This repeat sequence has been shown to trigger NGD in *S. cerevisiae* by causing strong stalling of ribosomal elongation (due to overall low abundance of tRNA$^{Arg(ICG)}$ compounded by inefficient decoding by the I:U wobble interaction) and ensuing ribosome collisions (*Letzring et al., 2010*; *Tesina et al., 2020*). A viral P2A sequence (*Brown and Ryan, 2010*; *Sharma et al., 2012*) was inserted between the GFP and *HIS3* open reading frames to decouple GFP levels from the protein decay induced by RQC factors in response to ribosome pausing on CGA codons. Importantly, knockout of *LTN1*, the major E3 ligase responsible for nascent peptide degradation by the RQC complex, did not increase the GFP/RFP ratio for our reporter (*Figure 1—figure supplement 1A*), demonstrating that nascent peptide decay does not impact the levels of GFP protein. As a result, for this construct, GFP levels serve as a proxy for reporter mRNA levels and translation initiation rates, allowing us to follow these activities in individual cells by flow cytometry. An RFP mRNA is produced from the same *GAL1-10* promoter in the reverse direction, allowing RFP fluorescence to be used to normalize for average transcription and metabolic changes within individual cells.

We performed a reporter-synthetic genetic array (R-SGA) screen (*Fillingham et al., 2009*) by introducing the GFP-OPT and GFP-CGA reporters into the 5,377 yeast strains contained in the Yeast Knockout Collection (*Giaever et al., 2002*). A total of 4222 deletion strains were successfully grown and tested with fluorimetry. We obtained GFP and RFP data for each deletion strain with these two reporters and calculated Z-scores of the GFP/RFP ratio for every strain on a per-plate basis, allowing comparison between the GFP-CGA screen and the previously published GFP-OPT screen (*Figure 1B*, *Figure 1—figure supplement 1B*; *D'Orazio et al., 2021b*). To identify genes contributing to NGD, we focused on knockout strains in which normalized GFP GFP-CGA reporter levels were significantly increased or decreased relative to our normalized GFP-OPT reporter levels (*Supplementary file 1*, *Figure 1B*, *Figure 1—figure supplement 1B*). Among the strongest hits from the screen were known NGD factors including *HEL2* and members of the RQT complex (*SLH1*, *CUE3*, and *RQT4*), all of which exhibited substantially decreased GFP reporter fluorescence compared to the wild-type control. This suggests that loss of these factors causes increased decay of the reporter mRNA. While initially counterintuitive, the observed modest decrease in mRNA levels suggests that decay can also be elicited by additional factors independent of Hel2 and the RQT complex, and that ribosome accumulation could increase the efficiency of this alternative pathway. These ideas were discussed in a recent review (*D'Orazio et al., 2021b*). To identify other potential factors, we looked among the strongest hits that increased GFP-CGA reporter levels and identified the ribosomal protein gene *ASC1* and the genes *SYH1* and *SMY2*, homologs of the mammalian NGD factors GIGYF1/2 which were previously reported to impact GFP-CGA reporter levels in yeast (*Hickey et al., 2020*). We also performed a similar R-SGA screen using a reporter identical to the GFP-CGA reporter, except with the CGA$_{12}$ repeat replaced by AAA$_{12}$; results from this screen showed broad overlap with the GFP-CGA reporter screen (*Figure 1—figure supplement 1C-D*) as anticipated based on the related stalling mechanisms of these sequences (*Tesina et al., 2020*; *Koutmou et al., 2015*).

Of the GFP-CGA reporter strains tested in the original screen, we selected 170 with the strongest increases or decreases in GFP levels (an absolute value of GFP-CGA Z-score greater than 2 and an absolute value of GFP-OPT Z-score less than 2) and individually validated them by flow cytometry to determine which backgrounds affected GFP-CGA reporter levels. As in the initial screen, *HEL2*, *SLH1*, and *RQT4* as well as *SYH1* and *SMY2* (*Supplementary file 2*, *Figure 1—figure supplement*

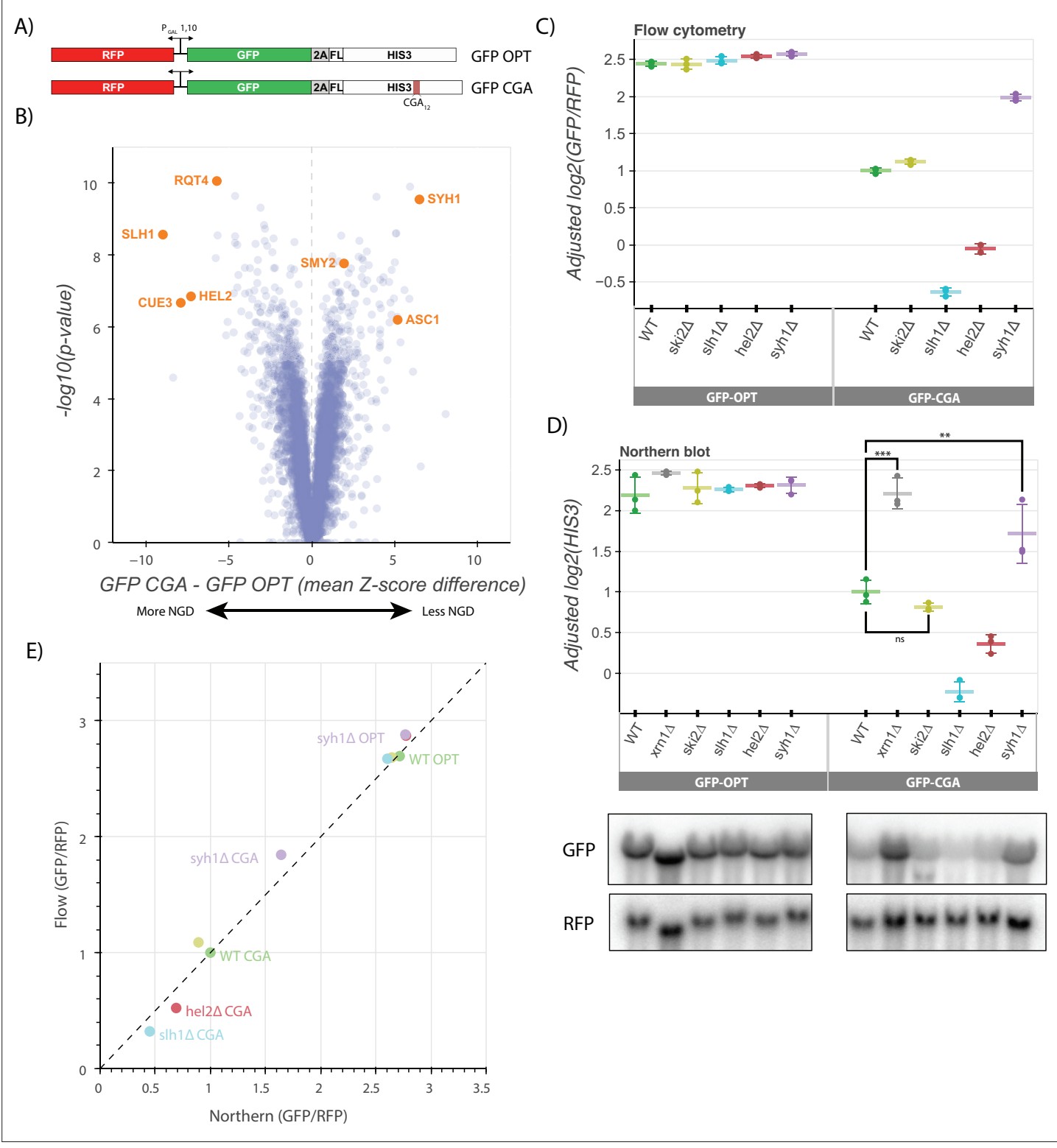

**Figure 1.** A genetic screen reveals factors that alter levels of an NGD reporter. (**A**) Diagrams of GFP-OPT and GFP-CGA reporters. Reporters are expressed from a bidirectional GAL promoter. GFP is separated from the HIS3 ORF by a P2A 'StopGo' sequence. The GFP-CGA reporter contains an insert of twelve CGA codons as a stalling sequence. (**B**) Volcano plot of data from the R-SGA screen. p-Values were calculated from a Fisher's t-test based on per plate GFP/RFP Z-score differences between the GFP-CGA and GFP-OPT screens. Positive Z-score differences indicate an increase of the GFP-CGA reporter relative to the GFP-OPT reporter and vice versa. Selected genes are labeled. (**C**) Flow cytometry analysis of GFP-OPT and GFP-

*Figure 1 continued on next page*

*Figure 1 continued*

CGA protein fluorescence reporter levels normalized to RFP fluorescence in several genetic backgrounds. All log2(GFP/RFP) levels are adjusted by normalizing to the mean of WT CGA and three replicates are plotted for each strain. Error bars indicate standard deviation. (**D**) Northern blot analysis of GFP-OPT and GFP-CGA GFP mRNA reporter levels quantified by probe hybridization and autoradiography. Three replicates are plotted for each background. All GFP values were fit to a linear mixed model to adjust plotted values for batch and loading effects and levels were normalized to the mean level of WT CGA. Tukey's honest significant difference test was performed and selected significance indicators are shown. *** indicates p-value <0.001, ** indicates p-vaue <0.01, ns indicates p-value >0.05. Error bars indicate standard deviation. Representative images of the northern blots for one replicate set are shown. (**E**) The unadjusted reporter levels from the experiments in panels C and D were normalized to RFP and further normalized to the mean WT CGA levels to allow comparison between flow cytometry and northern blot results.

The online version of this article includes the following source data and figure supplement(s) for figure 1:

**Source data 1.** Raw and labeled northern blot source images for *Figure 1D*.

**Source data 2.** Raw and labeled northern blot source images for *Figure 1C*.

**Figure supplement 1.** Analysis of R-SGA screens and reporters.

**Figure supplement 1—source data 1.** Flow cytometry source data for *Figure 1—figure supplement 1A*.

*1E*) strongly impacted GFP levels, validating our screen results and providing confidence for further mechanistic analysis. Of the 170 validated strains, 18 had statistically significant alterations in GFP-CGA reporter levels in our validation assay (*Supplementary file 3*), allowing them to be considered high-confidence hits. Based on manual inspection of gene annotations, these genes fell broadly into 4 categories: ubiquitin metabolism (*UBC4*, *DOA1*, *HUL5*, *SGF73*), tRNA modification (*NCS6*, *YNL120C*, *ELP6*), mitochondrial proteins (*MMR1*, *HMI1*, *SWS2*, *LPD1*), and known RNA decay factors and their homologs (*RQT4*, *EST1*, *SYH1*). A gene set enrichment analysis (GSEA) provides an analysis of these significant hits (*Supplementary file 4*, *Figure 1—figure supplement 1F*).

## Functional assays capture effects of NGD factors

To further explore the hits from our genome-wide screen, we deleted genes of interest de novo to verify the observed effects in a clean knockout background where we could further explore the mechanism of repression for the GFP-CGA reporter. We integrated the GFP-OPT and GFP-CGA reporters at the *ADE2* locus in yeast strains with deletions of *SYH1* and other factors including *HEL2*, *SLH1*, *SKI2*, and *XRN1*. While our screen identified two homologs of mammalian GIGYF1/2, our earlier studies had shown that stronger effects were associated with deletion of *SYH1* than with *SMY2* (*Hickey et al., 2020*); therefore for simplicity, we focused here on the *SYH1* deletion strain. In flow cytometry experiments, we found that the GFP/RFP fluorescence ratios for the GFP-OPT reporter were not strongly affected by deletion of *HEL2*, *SLH1*, *SKI2*, or *SYH1* (*Figure 1C*). In the wild-type strain, GFP-CGA reporter levels were reduced ~threefold in comparison to GFP-OPT as expected for a reporter subject to active NGD. Consistent with the screen results, GFP-CGA reporter levels were further reduced by about twofold in the *slh1Δ* and *hel2Δ* strains, while the GFP-CGA reporter levels were increased by ~1.7-fold in the *syh1Δ* strain (*Figure 1C*); in contrast, knockout of the *SKI2* gene had no effect on GFP-CGA reporter levels.

We next looked directly at mRNA levels in these strains using northern blotting to ask whether GFP levels reflect the mRNA levels. As seen in the flow cytometry assay, GFP-OPT mRNA reporter levels were unaltered across all the deletions strains and the GFP-CGA mRNA reporter level was reduced relative to the GFP-OPT mRNA reporter level in the wild-type strain (*Figure 1D*; Statistical significance for pairwise comparisons in all figures is calculated in *Supplementary file 5*). In agreement with the flow cytometry data and previous literature, *ski2Δ* had no effect on GFP-CGA reporter levels, whereas *xrn1Δ* strongly rescued reporter levels (*D'Orazio et al., 2019*; *Simms et al., 2018*). These results are consistent with the long established importance of Xrn1 in general mRNA decay (*Muhlrad et al., 1994*) and its critical role in NGD (*D'Orazio et al., 2019*). Other patterns observed in the flow cytometry data were recapitulated here as well: reporter levels were decreased to a similar extent in the *hel2Δ* and *slh1Δ* strains as in flow cytometry and *syh1Δ* significantly rescued mRNA reporter levels. The overall correlation between the flow data and the northern data for the GFP-CGA and the GFP-OPT reporters provided strong support for their use in an exploration of NGD (*Figure 1E*). This focused analysis of protein and RNA reporter levels in these different strains validates results from the

initial screen and indicates that the GFP-CGA reporter is being strongly regulated by canonical NGD machinery.

## Syh1 comprises an additional NGD pathway in yeast

Since screens in this study and our previous work (*D'Orazio et al., 2019*) implicated *HEL2*, *CUE2*, and *SLH1* in altering GFP-CGA reporter levels, we asked how knockouts of these factors would alter the reporter mRNA levels in an *syh1Δ* background to get some information about epistasis. Importantly, our previous work and others have demonstrated that the endonuclease activity of Cue2 is dependent on Hel2 (*D'Orazio et al., 2019*; *Ikeuchi et al., 2019*), enabling the inactivation of endonucleolytic NGD either by knockout of *CUE2* directly, or by knockout of *HEL2*. In order to compare more systematically the effects of these knockouts, we turned to a simpler set of reporters expressed from plasmids and containing problematic sequences within the *HIS3* gene but lacking the upstream GFP ORF and the P2A sequence (*Figure 2A*). The P2A sequence, in particular, has been shown to induce ribosome collisions during translation thus occasionally confounding interpretations (C. C.-C. *Wu et al., 2020*). The minOPT reporter is a codon optimized N-terminally FLAG-tagged *HIS3* sequence expressed under a *GAL* promoter; the minCGA reporter is identical to the minOPT reporter except it includes twelve CGA repeats in the same codon position within *HIS3* as in the GFP-CGA reporter used in screening.

To recapitulate our previous findings with the new minCGA stalling reporter, we used northern blotting to assess the steady-state levels of the reporter mRNA (*Figure 2B–C*). Although the *hel2Δ* strain decreases GFP-CGA reporter levels, it does not have the same effect on minCGA reporter levels; this discrepancy may arise from ribosome-stalling effects arising from the P2A sequence or from differences in levels of expression and ribosome loading. Importantly, although the loss of *CUE2* alone has little effect on the minCGA reporter, the deletion of *SLH1* results in a decrease in the amount of full-length mRNA and a corresponding accumulation of a 3' reporter fragment (*Figure 2B*, *HIS3* probe bottom band; *Figure 2C*, *Figure 2—figure supplement 1A*): these observations are wholly consistent with our previous studies that established that NGD proceeds through Xrn1 under normal circumstances, whereas the endonuclease Cue2 plays a more important role in the absence of *SLH1* (*D'Orazio et al., 2019*). In this paradigm, Cue2 acts as a failsafe that only cleaves mRNAs when stalled ribosomes accumulate in the *SLH1* deletion background. In order to investigate the effect of *SYH1* deletion on the minCGA reporter, we assessed reporter levels in an *syh1Δ* strain as well as a strains deleted for the *SYH1* paralog *SMY2*. While neither knockout individually rescued reporter levels to a statistically significant threshold, combining these knockouts did significantly increase minCGA levels over WT (*Figure 2—figure supplement 1B*). These observations of the role of Syh1 and Smy2 in decay are consistent with previous observations for the GFP-CGA reporter (*Hickey et al., 2020*) and suggest some functional overlap between Syh1 and its paralog Smy2.

To further probe the role of Syh1, we created double knockout strains with *syh1Δ* and other relevant factors and then performed northern blot analysis to follow the levels of the minCGA reporter mRNA. Strikingly, we observe that the *hel2Δsyh1Δ* strain shows a complete rescue of the minCGA mRNA to the levels of the minOPT reporter (*Figure 2C*); a similar strong rescue was seen with the GFP-CGA reporter (*Figure 2—figure supplement 1C*). Interestingly, the strong rescue phenotype observed in the *hel2Δsyh1Δ* strain is not recapitulated in the *hel2Δsmy2Δ* strain, though there is a small additional effect of *SMY2* deletion in all backgrounds (*Figure 2—figure supplement 1B*). We also observed significantly higher levels of minCGA reporter mRNA levels in the *cue2Δsyh1Δ* background, arguing that Cue2 cleavage (which requires Hel2 activity) is responsible for degrading mRNA in the absence of Syh1. Deletion of *SLH1* had little to no additive effect in the *cue2Δsyh1Δ* strain. These results suggest that Syh1-dependent (and to some extent Smy2-dependent) decay is a parallel pathway that contributes to NGD but that Cue2 plays a major compensatory role in decay when Slh1 or Syh1 activity is impaired.

## Syh1 is recruited to ribosome collisions in yeast by a mechanism distinct from its mammalian homolog

Several recent studies argued that mammalian GIGYF2 is recruited to collided ribosomes by the factor EDF1 (*Sinha et al., 2020*; *Juszkiewicz et al., 2020*). *S. cerevisiae* has a homolog of EDF1 known as Mbf1, a protein previously implicated in ribosome-mediated quality control pathways (*Hendrick*

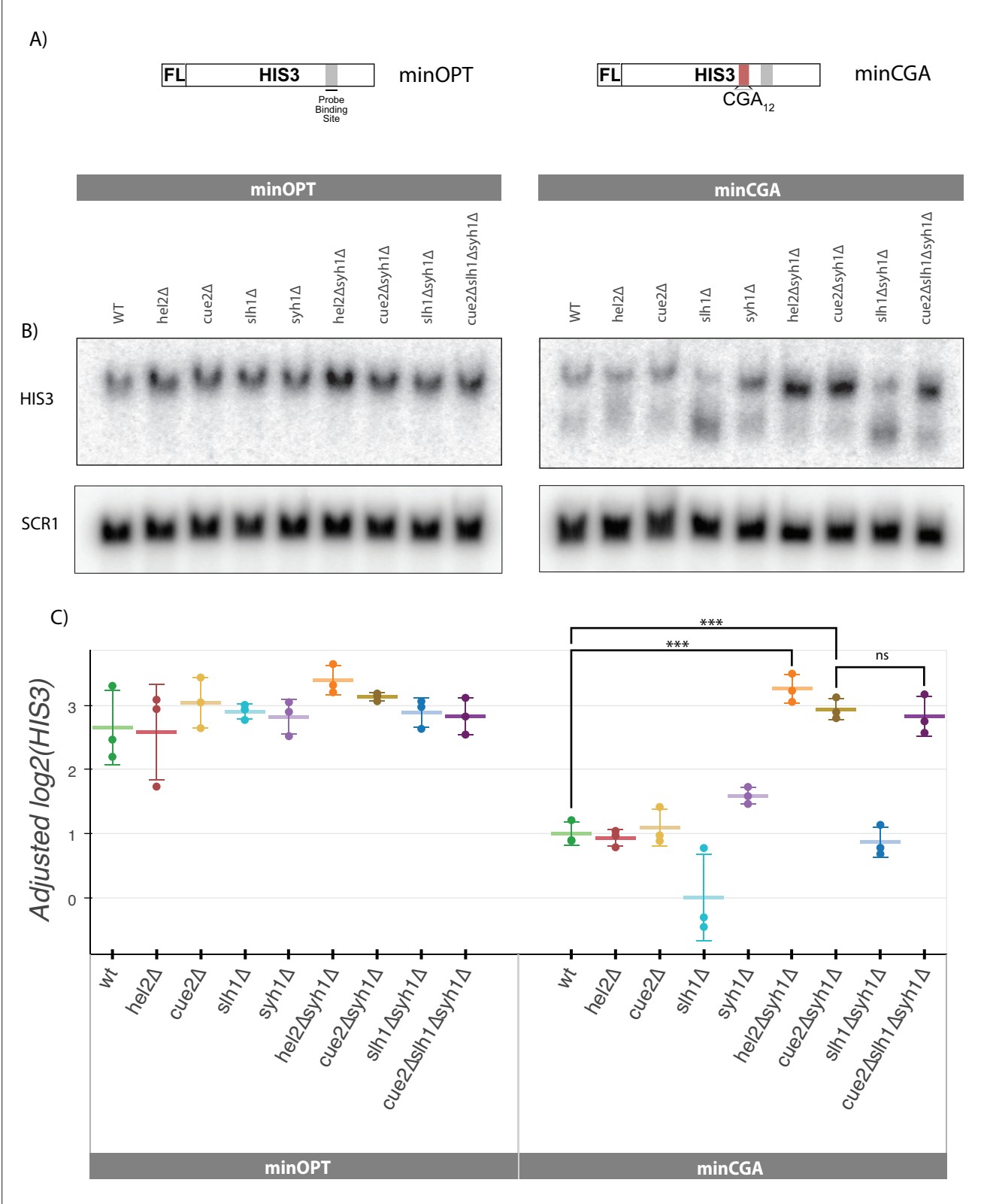

**Figure 2.** Syh1 is a critical mRNA decay factor for an NGD reporter. (**A**) Diagrams of minOPT and minCGA reporters. Reporters are expressed from a GAL promoter and contain only a FLAG tag (FL) and HIS3 ORF. The minOPT reporter contains a fully-optimized ORF, the minCGA reporter contains a $CGA_{12}$ stalling sequence within the ORF as indicated by red shading. (**B**) Autoradiograph of northern blot of one representative replicate set of reporter mRNA levels for strains and probes as indicated. (**C**) Northern blot quantification of minOPT and minCGA HIS3 mRNA reporter levels quantified by

*Figure 2 continued on next page*

*Figure 2 continued*

probe hybridization and autoradiography in yeast strains containing various NGD factor knockouts. Three replicates are plotted. All HIS3 values were fit to a linear mixed model to adjust plotted values for batch and loading effects and levels were normalized to the mean level of WT minCGA. Tukey's honest significant difference test was performed and selected significance indicators are shown. *** indicates p-value <0.001, ns indicates p-value >0.05. Error bars indicate standard deviation.

The online version of this article includes the following source data and figure supplement(s) for figure 2:

**Source data 1.** Raw and labeled northern blot source images and flow cytometry data for *Figure 2—figure supplement 1*.

**Source data 2.** Raw and labeled northern blot source images for *Figure 2B,C* and *Figure 2—figure supplement 1A*.

**Figure supplement 1.** Analysis of factors affecting NGD.

*et al., 2001*; *Wang et al., 2018*) that has been shown to interact with collided ribosomes in vitro (*Sinha et al., 2020*; *Pochopien et al., 2021*). Using tagged Syh1-TAP and Smy2-TAP, we performed affinity purification-mass spectrometry (AP-MS) to search for binding interactions that could help explain the mechanism of action of Syh1/Smy2. While our AP-MS data show strong enrichment of small and large subunit ribosomal proteins as well as components of the eIF3 complex, indicating a connection to translation, Mbf1 was not identified as bound to Syh1-TAP or Smy2-TAP (*Figure 2—figure supplement 1D*; *Supplementary file 6*).

To more systematically explore connections to *MBF1*, we looked at minCGA reporter levels in *MBF1* knockout strains. If Syh1 is recruited to ribosome collisions solely by Mbf1, we would expect to observe identical reporter levels in *MBF1* and *SYH1* deletion strains; our data initially supported this idea, with similar modest increases in minCGA reporter levels in both the *mbf1Δ* and *syh1Δ* strains (*Figure 2—figure supplement 1E*). However, we also wished to test whether the strong increase observed in the *hel2Δsyh1Δ* strain was recapitulated in the *hel2Δmbf1Δ* strain and surprisingly found that the *hel2Δsyh1Δ* strain and the *hel2Δmbf1Δ* strain did not phenocopy one another (*Figure 2—figure supplement 1E*). These data suggest that Syh1-dependent NGD is still active in the *hel2Δmbf1Δ* strain and thus that Mbf1 is not critical for RNA decay mediated by Syh1. Similarly, we examined GFP-CGA reporter expression in yeast strains lacking Eap1, an Syh1-associated translation repressor proposed to have a similar mechanism of action to the GIGYF2-associated mammalian protein 4EHP (*Sezen et al., 2009*; *Cosentino et al., 2000*; *Morita et al., 2012*; *Peter et al., 2019*). Again, GFP-CGA reporter levels are unaffected in the *eap1Δ* strain relative to WT and in the *eap1Δ* strain relative to *hel2Δeap1Δ* (*Figure 2—figure supplement 1F*); these data indicate that the Syh1-mediated loss in GFP signal occurs independently of Eap1. Additionally, Eap1 was not among the proteins identified in AP-MS (*Figure 2—figure supplement 1D*). These data together suggest that Syh1 recruitment and function in *S. cerevisiae* differs substantially from that observed for GIGYF1/2 in mammalian cells.

## COMD does not require canonical NGD factors

Given that the GFP-CGA and minCGA reporters simply contain a stretch of highly non-optimal codons, we wondered whether the same set of factors might similarly regulate ORF sequences containing more widely distributed non-optimal codons. We tested this possibility first by developing a reporter similar to that used for the NGD screen with an N-terminal GFP, an internal P2A sequence, and a downstream *HIS3* gene with an internal stretch of 129 codons synonymously re-coded as non-optimal (GFP-NONOPT; *Figure 3A*). As expected, the GFP-NONOPT reporter exhibited substantially diminished GFP levels compared to the GFP-OPT reporter and thus provided a starting point for subsequent analysis (*Figure 3B*).

To identify potential contributing factors to COMD, we performed an R-SGA screen (as above) with the GFP-NONOPT reporter (*Figure 3C*) and again compared these results to our GFP-OPT screen results. We first performed a GSEA analysis to identify categories of genes that may be involved in the regulation of nonoptimal mRNAs (*Figure 3—figure supplement 1A*) and note the enrichment of many categories involving cellular homeostasis and metabolism, including RNA metabolic processes. Strikingly, none of the known NGD factors that we had identified in the previous GFP-CGA screen emerged. These data provide a first indication that NGD mRNAs are regulated very differently from non-optimal mRNAs at the molecular level. Interestingly, factors previously implicated in stabilizing

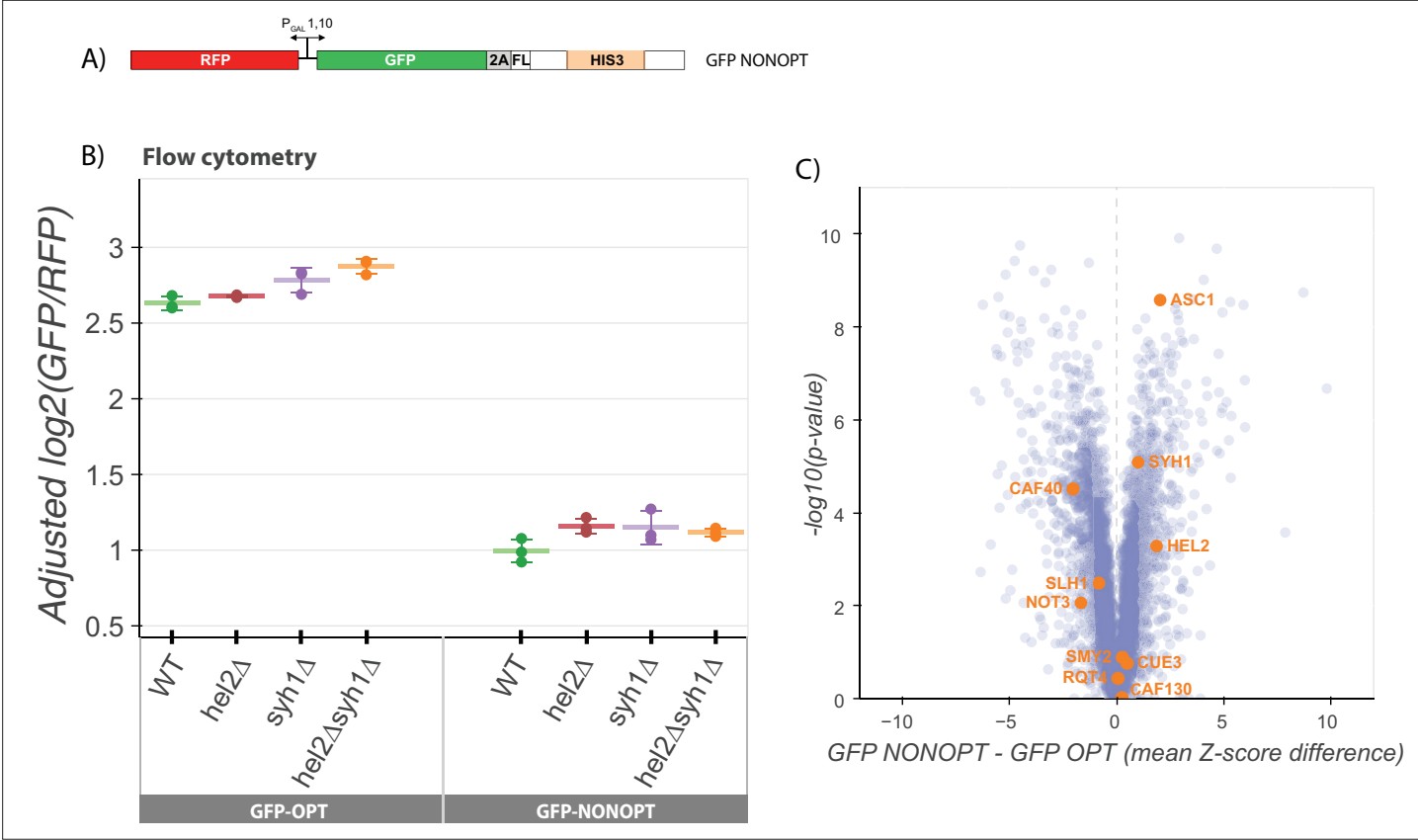

**Figure 3.** NGD factors do not alter levels of a COMD reporter. (**A**) Diagram of the GFP-NONOPT reporter for R-SGA screening and flow cytometry. Reporter is expressed from a bidirectional GAL promoter. GFP is separated from the HIS3 ORF by a P2A 'StopGo' sequence followed by a FLAG tag (FL). A portion of the HIS3 ORF is recoded as synonymous codons with low optimality. (**B**) Flow cytometry analysis of GFP-OPT and GFP-NONOPT GFP protein fluorescence reporter levels normalized to RFP fluorescence in yeast strains containing knockouts of HEL2 and SYH1 individually and in combination. All log2(GFP/RFP) levels are adjusted by normalizing to the mean of WT CGA and three replicates are plotted for each strain. Error bars indicate standard deviation. (**C**) Volcano plot of the GFP-NONOPT R-SGA screen p-values were calculated from a Fisher's t-test based on per plate GFP/RFP Z-score differences between the GFP-NONOPT and GFP-OPT screens. Positive Z-score differences indicate an increase of the GFP-NONOPT reporter relative to the GFP-OPT reporter and vice versa. Selected genes are labeled.

The online version of this article includes the following source data and figure supplement(s) for figure 3:

**Source data 1.** Flow cytometry source data for *Figure 3B*.

**Figure supplement 1.** Nonoptimal reporter screen results and comparison with other screens.

non-optimally coded mRNAs (*Webster et al., 2018*; *Radhakrishnan et al., 2016*; *Buschauer et al., 2020*) were also not among the deletion strains that revealed increases in GFP-NONOPT reporter levels (*Figure 3C*). While some of these strains are not present in the deletion strain collection due to their severe growth defects (*dhh1Δ* and *not4/5Δ*), deletions of several other members of the Ccr4-Not deadenylase complex were present in the screen including *caf40Δ*, *caf130Δ*, and *not3Δ*; among these, GFP-NONOPT reporter levels were modestly increased only in the *caf40Δ* strain.

Consistent with the results from the R-SGA screen, knockouts of the major factors implicated in our analysis of NGD (*hel2Δ* or *syh1Δ*) had no effect on GFP-NONOPT reporter expression levels (*Figure 3B*). We also compared the genes with high absolute Z-scores in our GFP-AAA and GFP-CGA reporter screens with the high absolute Z-score genes from the NON reporter screen (*Figure 3—figure supplement 1C-D*). We observe much less overlap in the top hits from either NGD screen with the NON screen than we observed between the two NGD screens (GFP-CGA and GFP-AAA). Together, these data suggest that there are fundamental differences in recognition by the decay machinery of overall non-optimal coding sequences and more problematic strong translational stalls.

## Comparing pathway responses between COMD- and NGD-triggering mRNAs

In order to compare more systematically the effects of mRNA sequences that trigger NGD or COMD, we turned again to the minOPT and minCGA reporters, with the addition of a third reporter called minNONOPT. This minNONOPT reporter encodes the same *HIS3* ORF recoded with highly non-optimal codons; importantly, there is a short stretch of sequence retained in the minNONOPT reporter that matches the minOPT reporter to allow a common oligonucleotide probe to be used in northern blots (*Figure 4A*). To evaluate these new reporters, we employed a northern-blot-based transcriptional shutoff assay to measure RNA half-lives as previously reported (*Radhakrishnan et al., 2016*) in various deletion strains including those implicated above in NGD as well as those previously implicated in COMD.

We first asked whether the half-lives of the minOPT reporter were affected by deletion of the various factors. As anticipated, neither of the factors implicated in regulation of NGD reporters (*HEL2* or *SYH1*) had any discernible effect on the stability of the minOPT reporter mRNA (*Figure 4B*, *Figure 4—figure supplement 1A-B*, *Supplementary file 7*). We next determined the effects of the same factors on stability of the minNONOPT mRNA reporter and found that deletion of the NGD factor genes *HEL2* and *SYH1* had no effect while *NOT5* deletion stabilized this mRNA by approximately 11-fold, consistent with earlier reports (*Buschauer et al., 2020*).

Lastly, we evaluated the effects of the same set of factors on minCGA mRNA stability. There we found that deletion of *HEL2* has no discernible impact on minCGA reporter half-life and deletion of *SYH1* increases half-life twofold, while deletion of both factors (*hel2Δsyh1Δ* strain) increases half-life sevenfold, consistent with results from our steady-state northern blots. These results additionally agree with steady-state measurements reported for the screening reporter constructs (*Figure 2—figure supplement 1C*). Finally, somewhat surprisingly, we find that deletion of *NOT5* increases minCGA reporter half-life (~twofold), suggesting features of general non-optimality for this reporter mRNA. Together, these data are broadly consistent with the existence of multiple modes for recognition of troubled elongating ribosomes, each recognized by a distinct set of factors.

## Distinct ribosome signatures on ORF stalling motifs

With the goal of connecting ribosome states to downstream consequences, we next employed ribosome footprint profiling (Ribo-seq) to characterize the positions and conformational states of elongating ribosomes on the minimal reporter mRNAs (*Ingolia et al., 2009*). To increase the resolution of the approach, both cycloheximide and tigecycline, two specific elongation inhibitors, were added to lysates to capture distinct rotational states of the ribosome during translation represented by two populations of ribosome protected fragment (RPF) lengths centered at approximately 21 and 28 nucleotides (*Wu et al., 2019b*). The 21-mer population captures ribosomes without A-site tRNAs (waiting to decode) while the 28-mer population captures ribosomes with filled A sites (waiting to translocate). Finally, to further increase the resolution of the study, we separately sequenced single-ribosome (monosome) footprints and those from nuclease-resistant disomes that represent the collided ribosome structure (*Guydosh and Green, 2014*).

We first looked at the distribution of RPFs along our reporter mRNAs. We see that ribosomes are relatively evenly distributed across the entire ORF in the minOPT reporter for both the 21-mer and 28-mer tracks (*Figure 5A*). As previously observed for the GFP-CGA reporter (*Sitron et al., 2017*; *D'Orazio et al., 2019*), we observed that the CGA stall region profoundly disrupts translation of the minCGA ORF. The density of monosome ribosome footprints (both 28- and 21-mers) downstream of the CGA stall is greatly reduced compared to the upstream region on this reporter, indicating that the CGA repeat region comprises a significant translational block (*Figure 5A*, center; *Figure 5—figure supplement 1B*, top). Importantly, there is a high density of 21-mers near the beginning of the CGA repeat region, consistent with previous reports that decoding of CGA codons is slow. Additionally, 21 and 28-mer peaks appear approximately one footprint length upstream of the CGA stall, extending backwards in a repeating pattern approximately every ribosome length as confirmed by autocorrelation analysis (*Figure 5—figure supplement 1A*, *Figure 5—figure supplement 1C* top row).

Disome profiling also reveals nuclease-resistant collided ribosome footprints accumulating in this region upstream of the CGA codons; and, autocorrelation analysis reveals that the pattern of 'stacked' disomes is periodic with peaks at approximately 30 nucleotide intervals (*Figure 5B*, center; *Figure 5C*,

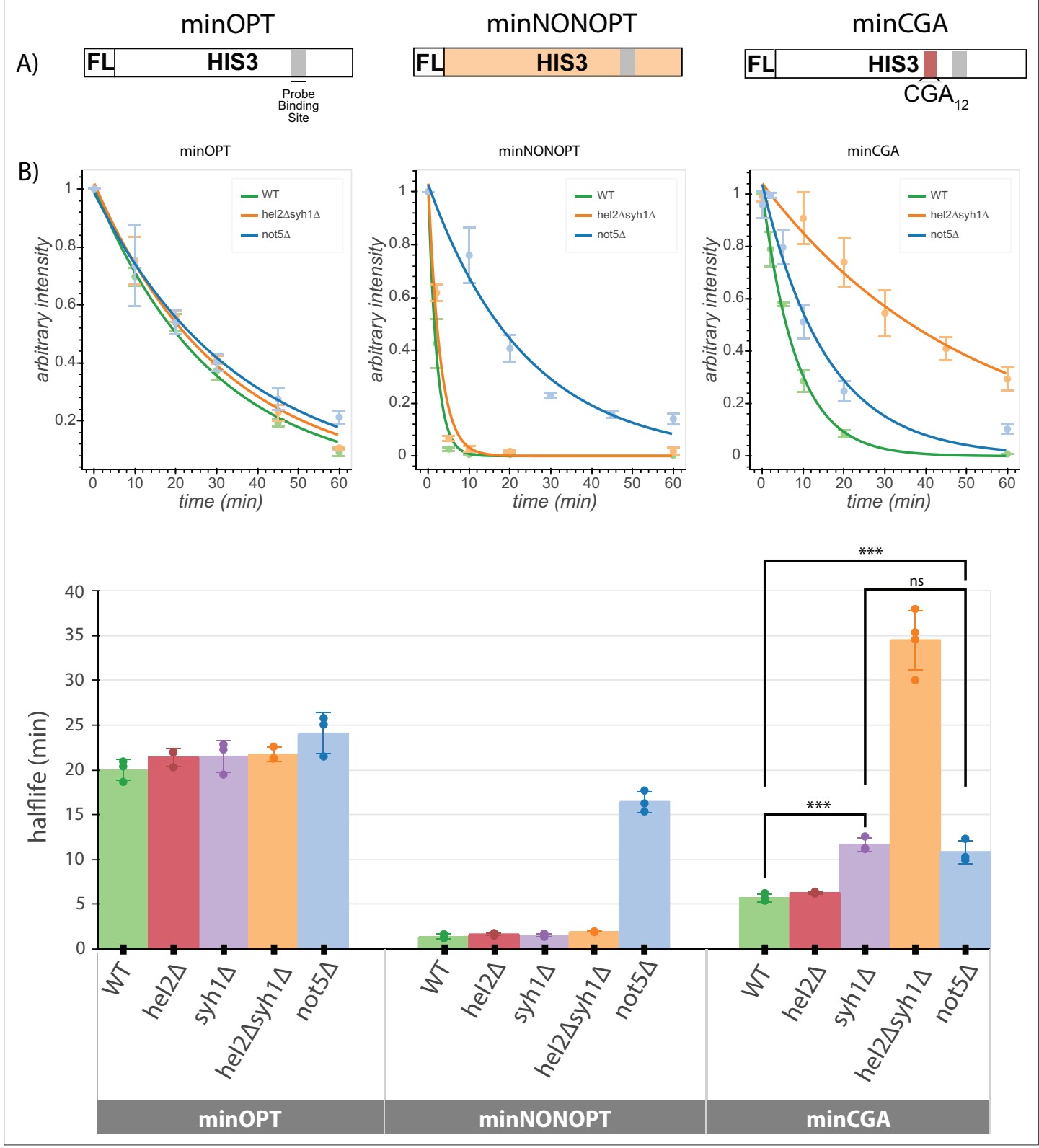

**Figure 4.** Minimal reporter mRNA half-lives are affected by COMD and NGD factor knockouts. (**A**) Diagram of the minOPT, minCGA, and minNONOPT reporters. Reporters are expressed from a GAL promoter and contain only a FLAG tag (FL) and HIS3 ORF. The minOPT reporter contains a fully-optimized ORF, the minCGA reporter contains a $CGA_{12}$ stalling sequence within the ORF (indicated by the red shaded region), and the minNONOPT reporter contains a HIS3 ORF fully recoded as synonymous non-optimal codons. All reporters share a common probe binding region for northern blot

*Figure 4 continued on next page*

*Figure 4 continued*

analysis (shaded in gray). (**B**) Reporter mRNA half-lives in distinct genetic backgrounds were measured following transcriptional shut down. Full-length reporter mRNA levels were normalized to endogenous SCR1 levels. Top, reporter mRNA decay curves measured by northern blot analysis after GAL promoter shutoff in different genetic backgrounds. A single exponential decay was fit to means of three or four replicates at each timepoint. Bottom, half-lives were calculated for replicates individually by fitting to a single exponential decay and averaged. Half-lives were fit to a linear model and Tukey's honest significant difference test was performed. Selected significance indicators are shown. *** indicates p-value <0.001, ns indicates p-value >0.05. All error bars indicate standard deviation.

The online version of this article includes the following source data and figure supplement(s) for figure 4:

**Source data 1.** Raw and labeled *not5Δ* strain northern blot source images for *Figure 4*.

**Source data 2.** Raw and labeled *syh1Δ* strain northern blot source images for *Figure 4*.

**Source data 3.** Raw and labeled wild-type strain northern blot source images for *Figure 4*.

**Source data 4.** Raw and labeled *hel2Δ* strain northern blot source images for *Figure 4*.

**Source data 5.** Raw and labeled *hel2Δsyh1Δ* strain northern blot source images for *Figure 4*.

**Figure supplement 1.** Galactose shutoff mRNA decay of reporters in knockout backgrounds.

red trace). In this analysis, disome RPM peaks on the minCGA reporter are dramatically increased (~10-fold) relative to the minOPT reporter, indicative of an accumulation of collided ribosomes (compare scales of minOPT and minCGA panels in *Figure 5B*). These data together suggest that a key signal for the recruitment of Syh1 and the NGD machinery is the collided ribosome.

To further explore a potential ribosomal basis for the activity of NGD factors on stalling reporter levels, we performed ribosome profiling on the minCGA reporter in *hel2Δ* and *syh1Δ* strains. First, consistent with previous observations (*Letzring et al., 2013*), deletion of *HEL2* increased ribosome read-through past the stall region relative to the wild-type strain (*Figure 5—figure supplement 1B*). Additionally, we noted from the disome profiling that the ordered periodicity in footprint distribution that is present in the WT strain is diminished in the *hel2Δ* strain as revealed by an autocorrelation analysis (*Figure 5C* gray trace, *Figure 5—figure supplement 1D*). The same diminished periodicity is observed in monosomes in the *hel2Δ* strain, whereas the ribosome periodicity is generally maintained in the *syh1Δ* strain (*Figure 5—figure supplement 1C*).

In contrast to the minCGA reporter, the minNONOPT reporter contains non-optimal codons distributed throughout the ORF; accordingly, monosome footprints are relatively evenly distributed across the ORF (*Figure 5A*, right panel). Disome footprints are distributed across the ORF as well, suggestive of stochastic short-lived collisions where the ribosome density does not accumulate. To allow for accurate comparison, we performed a more detailed analysis of 21-mer/28-mer ratios for the reporter after excluding the identical oligonucleotide probe region for the two reporters as well as regions around the start and stop codons that can be sensitive to library preparation and variable within ribosome profiling data sets (*O'Connor et al., 2016*). Importantly, we normalized these data from the reporter ORF to global ORF 21-mer/28-mer ratios within each dataset to account for differing RNase digestion efficiencies in the library preparations. Our data reveal an increased 21-mer/28-mer ratio for the minNONOPT relative to the minOPT reporter; as 21-mer RPFs report on empty A sites on the elongating ribosomes, these data are consistent with an enrichment of ribosomes undergoing slow decoding of tRNAs during elongation on non-optimal codons (*Figure 5D*). These data suggest that a key signal for the recruitment of the COMD machinery (including Not5) is the accumulation of slowly decoding ribosomes.

Given the abundance of 21-mer RPFs on the minNONOPT reporter and the strong decay activity of Not5 on this reporter, we wondered whether the activity of Not5 in reducing minCGA reporter half-life (*Figure 4B*) could be explained by accumulation of 21-mer RPFs. Interestingly, a deeper analysis of the distribution of multiply aligned monosome footprint reads *within* the CGA repeat region of the minCGA reporter revealed that the 21mer/28mer read ratio is greatly enriched compared to the ratio on all genes (*Figure 5E*). We suggest that this preponderance of slow (21mer RPFs) elongating monosomes in the CGA repeat region might explain the partial sensitivity of the NGD reporter to Not5-mediated destabilization. Since Not5 recognizes ribosomes with open A and E sites, it may be able to bind those ribosomes that continue translating within the $CGA_{12}$ region to elicit decay of the reporter.

Taken together, these experiments provide compelling data to rationalize the differing responses of our reporters to various gene deletions. The accumulation of distinct ribosome signals—either

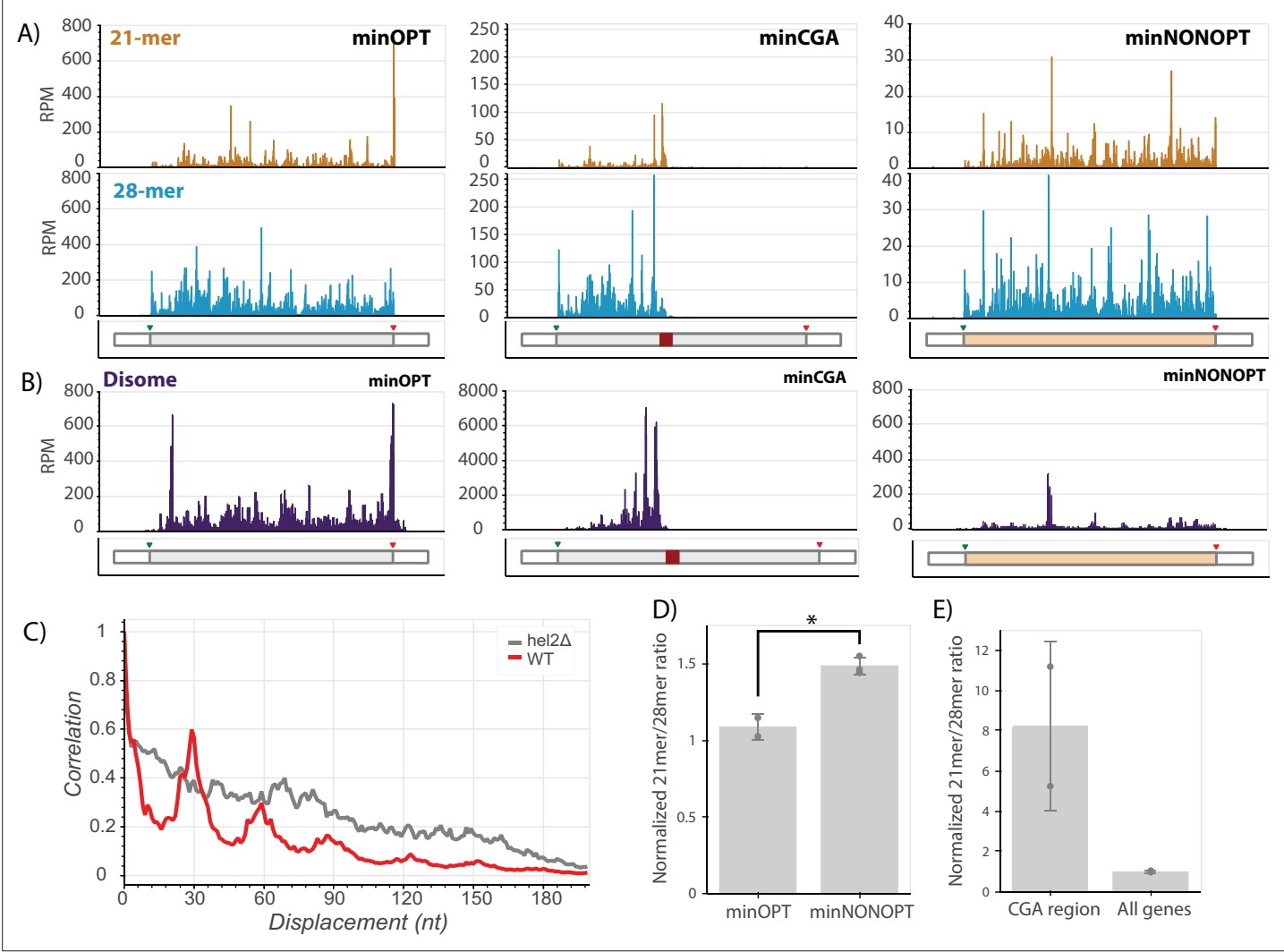

**Figure 5.** Ribosome profiling reveals translational states that trigger NGD and COMD. (**A**) Gene diagrams show reads per million (RPM) at every position of the indicated reporters: minOPT (left), minCGA (center), minNONOPT (right). Top, 21-mer reads are plotted using read lengths 19–26 inclusive. Bottom, 28-mer reads are plotted using read lengths 27–36 inclusive. Beneath each plot, diagrams of the indicated reporter show the locations of important features. (**B**) Read RPMs from disome footprint profiling are shown for each position of the minOPT (top) and minNONOPT (bottom) reporters. (**C**) Autocorrelation of the disome footprint RPMs in the region upstream of the CGA repeats for WT minCGA and hel2Δ minCGA samples. (**D**) Normalized 21-mer to 28-mer ratios of two replicates for the minOPT and minNONOPT reporters. Ratios exclude regions near the start and end of the ORF and the common probe binding region and are normalized to the 21-mer to 28-mer ratio of all genes to control for differences in digestion efficiency between libraries. Error bars indicate standard deviation. Significance was tested with a t-test, * indicates a p-value <0.05. (**E**) Normalized 21-mer to 28-mer ratios of two replicates for multiply aligned reads within the CGA region of the minCGA reporter. Ratios are normalized to the average of 21-mer to 28-mer ratios for all genes. Error bars indicate standard deviation.

The online version of this article includes the following figure supplement(s) for figure 5:

**Figure supplement 1.** Ribosome profiling analysis.

nuclease-resistant collided disomes or slowly decoding ribosomes with empty A sites—acts as a strong indicator for which factors will lead to mRNA destabilization; Syh1 and Hel2 respond to the accumulation of terminally stalled, collided ribosomes, while Not5 responds to slowly decoding ribosomes with open A and E sites.

## Discussion

In this study, we use carefully designed reporter mRNAs to study translation-coupled mRNA decay pathways in *S. cerevisiae*. Using R-SGA screening with a reporter mRNA containing iterated CGA codons, we identified and validated a set of genes that contribute to no-go decay (NGD). Subsequent analysis allowed us to compare the mechanisms of the pathways that regulate decay of mRNAs with either highly problematic (NGD) or slowly decoded sequences (COMD). We find that an independent pathway of NGD is driven primarily by the actions of the GIGYF1/2-homologous proteins Syh1 and Smy2; in contrast, the Syh1/Smy2-dependent pathway has no discernible impact on the stability of non-optimal mRNA sequences. We show that the previously defined COMD factor Not5 contributes modestly to decay of the NGD reporter and very strongly to decay of non-optimally coded mRNAs. Finally, we connect these distinct molecular decay profiles with ribosome states using ribosome profiling, showing that the presence of colliding ribosomes (disomes) is correlated with targeting via NGD while the presence of slow ribosomes (monosomes with empty A and E sites) is correlated with targeting via COMD.

Our assays using the NGD reporters reveal the interplay between Hel2, its dependent NGD factors, and Syh1/Smy2 in responding to ribosome collisions. First, we found that combining deletion of *SYH1* and its paralog *SMY2* led to modest stabilization of NGD reporter mRNAs and deletion of *CUE2* and *SYH1* together led to very potent stabilization (*Figure 2C*). In light of previous work establishing the relationship between Hel2, Cue2, and Slh1 (*D'Orazio et al., 2019*), we interpret the mRNA stability data as follows: (1) in the wild-type strain, the NGD reporters are destabilized because colliding ribosomes lead to the recruitment of Syh1/Smy2 and elicit mRNA destabilization while Hel2-mediated ubiquitination licenses Slh1-mediated ribosome clearing and Cue2-mediated endonucleolytic cleavage (2) in the *syh1Δsmy2Δ* strain, NGD reporter levels are only partially rescued because Cue2-mediated endonucleolytic decay can still trigger NGD; and (3) in the *hel2Δsyh1Δ* and *cue2Δsyh1Δ* strains, NGD reporters are strongly stabilized because both Syh1-dependent and Cue2-dependent decay pathways are impaired. According to this view, there are two distinct pathways for NGD that can compensate for each other in various conditions. This model gives new context to the pathways that respond to elongation stalls and emphasizes the importance of Syh1, and to a lesser extent Smy2, as distinct effectors of NGD. These data further support our previously discovered mechanism in which the ubiquitination activity of Hel2 triggers RQT-mediated ribosome rescue through Slh1, resorting to Cue2-dependent endonucleolytic cleavage when mechanisms to resolve ribosome collisions are overwhelmed. The synergistic activities of Syh1/Smy2- and Hel2-triggered decay form the basis of a robust cellular system for targeting problematic mRNAs for destruction (*Figure 6*).

This interplay was further explored in our Ribo-seq experiments. For the minCGA reporter, we observed an ordered, periodic pattern of monosome and disome footprints upstream of the CGA region in the WT strain, and this pattern was substantially disrupted in the *hel2Δ* strain. Further, we observe a larger proportion of 'escaping' monosome footprints downstream of the CGA region in the *hel2Δ* strains. Given the role of Hel2 in recognizing collided ribosome and promoting clearance by RQT, we interpret these data to mean that the activity of Hel2 on collided ribosomes in part stabilizes the collided ribosome interface in such a way that recognition and clearance by Slh1 is facilitated (*Meydan and Guydosh, 2020*). These observations are distinct from the maintenance of wild-type-like periodicity when *SYH1* is knocked out (*Figure 5—figure supplement 1C*).

The next set of questions focused on how Syh1 (and its homolog Smy2) is recruited to problematic mRNAs and how it triggers translational repression or mRNA decay. Recent work in mammalian cells investigating the mechanism of recruitment of GIGYF2 to NGD-targeted mRNAs yielded two competing models: one in which ZNF598 (a mammalian *HEL2* homolog) acts to recruit GIGYF2 (*Hickey et al., 2020*) and one in which EDF1 acts to recruit GIGYF2 (*Sinha et al., 2020*; *Juszkiewicz et al., 2020*). We tested both models using our yeast reporter system. First, we observed strong stabilization of the GFP-CGA reporter mRNA levels when *SYH1* is deleted in a *hel2Δ* background (comparing *hel2Δ* to *hel2Δsyh1Δ*) thus establishing that Hel2 function is not necessary for Syh1 function in yeast. Second, we did not observe the same strong rescue of minCGA reporter mRNA levels in a *hel2Δmbf1Δ* strain as we did in a *hel2Δsyh1Δ* strain, suggesting that Syh1 function is also not dependent on Mbf1. Although Mbf1 does bind collided disomes in yeast (*Sinha et al., 2020*; *Pochopien et al., 2021*), these data suggest that Mbf1 has a limited impact on NGD per se in this system. A

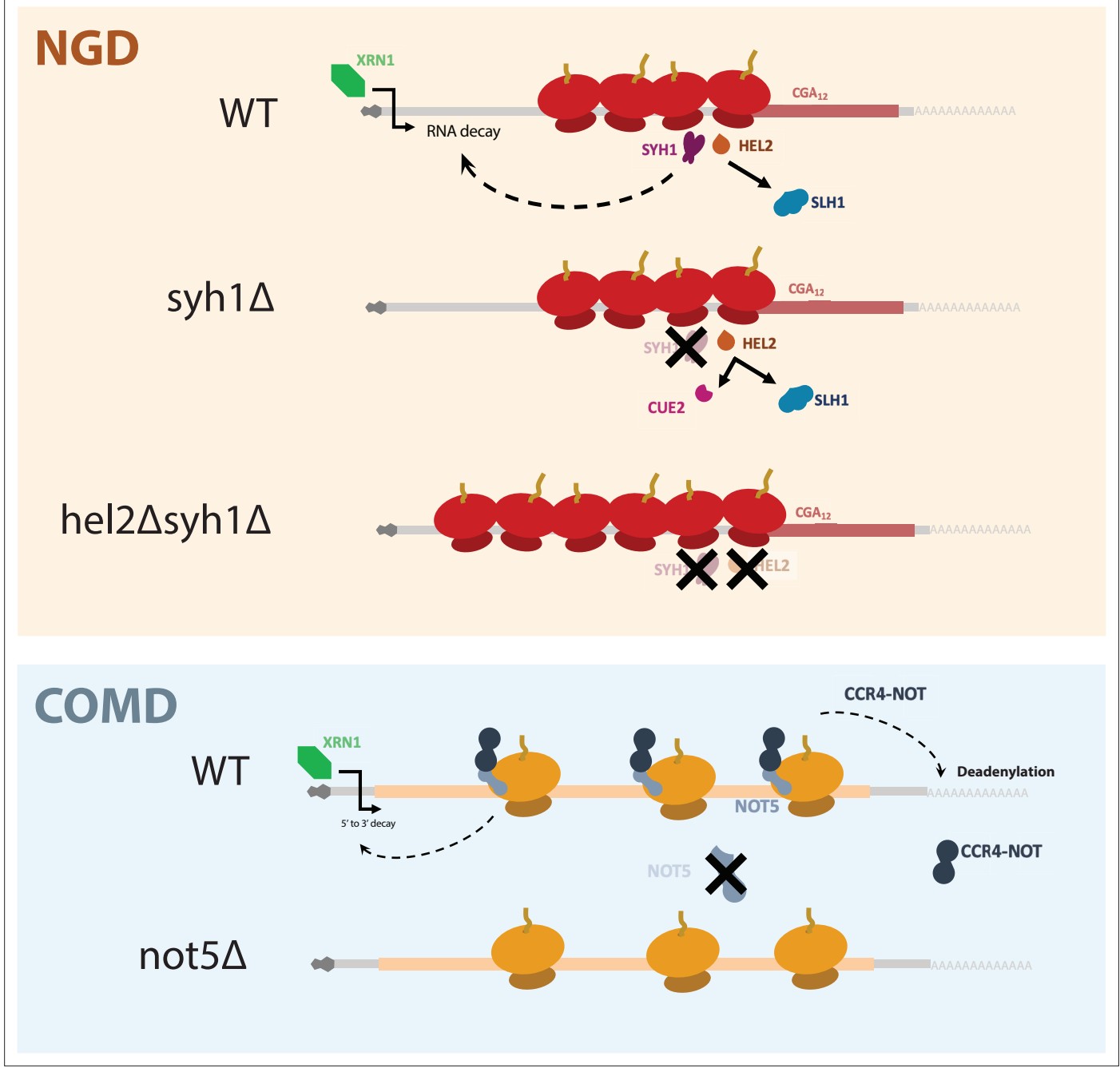

**Figure 6.** A model for NGD and COMD. In NGD, Syh1 responds to collided ribosomes, connecting severe translational blocks to mRNA decay. Loss of Syh1 results in activation of Hel2-dependent endonucleolytic NGD by Cue2. Loss of Syh1 and Hel2 causes increased reporter accumulation by blocking both exonucleolytic and endonucleolytic RNA decay pathways. In COMD, Not5 senses slow ribosomes on non-optimal codons and recruits the Ccr4-Not complex, causing deadenylation, decapping, and 5' to 3' decay. Loss of Not5 stabilizes non-optimal mRNAs.

previous study that employed bioID mass spectrometry using Syh1 as bait (*Opitz et al., 2017*) raises the possibility that Syh1 interacts directly with Asc1, a ribosomal protein known to be important for NGD in yeast (*Kuroha et al., 2010*; *Letzring et al., 2013*; *Brandman et al., 2012*), and a top effector of NGD in our genetic screen (*Figure 1A*). Further work exploring the nature of the physical interactions between Syh1/Smy2 and ribosomal collisions is necessary to make stronger conclusions about the mechanism of Syh1/Smy2-mediated NGD.

We also considered a role for other factors in facilitating Syh1-mediated decay, including the potential eIF4E2 homolog, EAP1, which did not affect reporter levels (*Figure 2—figure supplement*

*1F*). Another candidate effector protein for Syh1 function is Dhh1 whose mammalian homolog DDX6 interacts with GIGYF1/2 and has been shown to facilitate translational repression (*Peter et al., 2019*; *Weber et al., 2020*). In yeast, however, there is scarce evidence for a Dhh1-Syh1/Smy2 complex (*Ergüden, 2019*) and the conserved DDX6 binding motif of GIGYF1/2 is absent in Syh1/Smy2 (*Figure 2—figure supplement 1G*). These data together raise the interesting possibility that Syh1 and Smy2 have a distinct mechanism of regulation in yeast that involves direct signaling of mRNA decay independent of translational repression. We note that the strong mRNA decay phenotype associated with Syh1/Smy2 function and the NGD reporters in yeast (here and in *Hickey et al., 2020*) is distinct from the translational repression phenotype associated with GIGYF2:4E2 function in mammalian systems (*Morita et al., 2012*; *Peter et al., 2019*).

Although NGD and COMD are both known to be co-translational mRNA decay processes, the distinct ribosome states that trigger these events have not been systematically explored. For example, it has not been clear whether ribosomes translating highly non-optimal CGA repeats (that efficiently trigger NGD) may also be recognized by Not5. Conversely, the possibility remained open that highly non-optimal reporter mRNAs cause ribosome collisions that activate NGD in addition to COMD.

Our analysis of the COMD reporters revealed responses only to factors previously implicated in this pathway (*Buschauer et al., 2020*). The half-lives of the minNONOPT reporter were not impacted by deletion of *SYH1, HEL2,* or *HEL2/SYH1* (*Figure 4B*) and the ribosome profiling data revealed a relatively even distribution of monosomes and low abundance of disome peaks. Additionally, we observed an enrichment of 21 nt RPFs over the coding region representing slowly elongating ribosomes (*Figure 5D*). These data are broadly consistent with the previously reported Not5-driven mechanism being responsible for COMD with essentially no contribution from colliding ribosomes and NGD. By contrast, the minCGA reporter was strongly stabilized by deletion of factors implicated in NGD (Syh1/Smy2 and Hel2) and weakly stabilized by deletion of factors implicated in COMD (Not5) (*Figure 4B*). The ribosome profiling data revealed abundant collided disomes at the CGA repeat sequences, but also an enrichment of 21 nt monosome RPFs in the actual CGA repeats reflecting ribosomes struggling their way through these difficult to decode sequences (*Figure 5B and E*). These data rationalize both the strong contribution of the NGD factors and the more modest contribution of the COMD factors to minCGA reporter stability.

Our study provides evidence for generally non-overlapping targets of NGD and COMD and ribosome states correlated with each pathway. While the NGD machinery, under the control of Syh1/Smy2 and Hel2, responds to specific defects in elongation due to stalled and collided ribosomes, the COMD machinery, under the control of Not5, surveys the pool of translating ribosomes for mRNAs on which there is overall slow translation. We speculate that the COMD pathway is a general one that regulates overall mRNA stability, independent of ribosome dysfunction, while the NGD pathway evolved to deal with more acute environmental disturbances such as UV or oxidative damage (*Yan et al., 2019*; C. C.-C. *Wu et al., 2020*). Future studies will better characterize the molecular mechanisms of these pathways and will provide new foundations for an understanding of the homeostasis of cellular translation and mRNA decay.

# Materials and methods
## Reporter cloning and reporter strain generation
Plasmids for GFP-OPT (pKD065), GFP-NONOPT (pKD064), and GFP-CGA (pKD080) reporters were cloned as described in *D'Orazio et al., 2019*. To generate stable, genomically integrated strains containing these reporters, 0.5–2 μg of plasmid was digested using StuI to produce an insertion cassette containing the RFP and GFP reporters plus a *MET17* gene for selection, all flanked by homology arms to the endogenous *ADE2* locus. Strains were then transformed by lithium acetate transformation to replace the *ADE2* gene as described below, with the difference of being plated directly onto selective media after transformation rather than a nonselective recovery plate. The minimal reporters minOPT and minNONOPT (plasmids pJC867 and pJC857, respectively) were a generous gift from Jeff Coller and cloned as described in *Radhakrishnan et al., 2016*. The CGA repeat stretch was introduced into the pJC867 plasmid by first isolating the plasmid backbone via digestion with PacI and AscI. Two PCR fragments making up the *HIS3* ORF were generated, one containing the first portion of the *HIS3* ORF and a the CGA repeat region in a primer overhang (primers AV_his3CGAupstr_fwd and

AV_his3CGAupstr_rev, see table of oligos), the other containing the downstream *HIS3* sequence and stop codon (primers AV_his3CGAdwnstrm_fw and AV_his3CGAdwnstrm_rv, see table of oligos). Both fragments were amplified off of pJC867 and inserted into the linearized pJC867 backbone using NEB Gibson Assembly Master Mix resulting in pAV_minCGA plasmid. Since these plasmids contained a URA3 selectable marker, reporters were introduced into various backgrounds by transformation of 0.5–1 µg plasmid as described below and all subsequent culture was performed in SC-URA media (plus additives appropriate to experiment) to retain the plasmid.

## R-SGA screening
### Screening procedure
Screening was performed as described in *D'Orazio et al., 2019*. Briefly, GPF-OPT, GFP-CGA, GFP-NONOPT, and GFP-AAA reporters were inserted into the Yeast Knockout Collection (*Giaever et al., 2002*) by mating and four replicate colonies were grown for each strain. Incubation times were increased during this process by 50–75% to account for decreased mating efficiency in our query strains. Cells were selected on appropriate media, then plated on 2% GAL/RAF media for expression and analyzed by imaging on a Typhoon FLA9500 imager.

### Screen data analysis
Data was analyzed as previously described in *D'Orazio et al., 2019* and *D'Orazio et al., 2021b*. GFP-OPT screen data used for normalization is the same data previously published in *D'Orazio et al., 2021b*. Briefly, median GFP and RFP values were extracted from colony images using specialized software (*Saeed et al., 2003*; *Wagih et al., 2013*) and outliers were excluded (border colonies and those <1500 or>6000 pixels). Average GFP and RFP values from all colonies were then converted to $\log_2$(GFP/RFP) ratios and LOESS normalized on each plate. Z-scores were calculated on a per-plate basis. Z-scores for volcano plots were calculated without prior LOESS normalization.

### Screen validation
The 170 Yeast Knockout Collection strains with integrated reporters that showed the greatest change in GFP/RFP for the GFP-CGA reporter (–2>GFP CGA Z-score >2) and lowest change for the GFP-OPT reporter (–2<GFP OPT Z-score <2) relative to a *his3Δ* control strain were selected and grown to saturation overnight in YP +2% galactose +2% raffinose media in deep-well 96-well plates. Cultures were diluted in triplicate to approximately $OD_{600}$ 0.1 in deep-well 96-well plates and grown to approximately $OD_{600}$ 0.4–0.6. A 10 µL aliquot of culture was then added to 190 µL PBS and flow cytometry was performed as described below in a Guava EasyCyte HT flow cytometer. p-Values were calculated by Fishers T-test.

## Yeast strain generation, culture, and harvesting
### Knockout strain generation
Knockout strains were created using the BY4741 (*MAT**a** his3Δ1 leu2Δ0 met15Δ0 ura3Δ0*) background as wild-type. DNA fragments containing 40–70 nt homology arms to the gene of interest were amplified by PCR using MX cassette plasmids as template (*McCusker, 2017*) and purified using a Zymo DNA Clean & Concentrator-5 kit. Yeast were then transformed using high-efficiency lithium acetate transformation (*Gietz and Schiestl, 2007*). Briefly, strains to be transformed were grown to saturation at 30 °C overnight in an appropriate medium (typically YPD, YPAD, or SC-Ura), then diluted to $OD_{600}$ 0.2 in 5 mL media. Meanwhile, the transformation mixture was prepared, consisting of 33% PEG 3350, 100 mM LiAc, 0.28 mg/mL boiled salmon sperm DNA, and 1–5 µg PCR product. When cultures reached $OD_{600}$ 0.4–0.6, they were harvested by centrifugation (3000xg, 5 min) and resuspended in the transformation mixture. Transformation mixtures were incubated at 42 °C (or 30 °C for *not5Δ* strains) on a thermomixer for 40–60 min, then centrifuged briefly to collect a yeast pellet, discarding the supernatant. Finally, yeast were resuspended in 200 µL water (or media for *not5Δ* strains), plated on an appropriate nonselective agar medium and incubated at 30 °C. Transformants were then streaked to single colonies on a fresh plate, and these colonies were tested for MX cassette insertion by PCR using Phire Plant Direct PCR Master Mix. Confirmed strains were later maintained as patches on selective agar medium.

## Growth conditions

Unless noted otherwise, yeast for steady-state reporter expression measurements (by flow cytometry or northern blot) and ribosome profiling were grown to saturation in an appropriate medium lacking glucose and containing 2% galactose and 2% raffinose. Cells were then diluted to $OD_{600}$ 0.1 and grown to $OD_{600}$ 0.4–0.65 before being harvested according to the requirements of the particular assay to be performed.

## Flow cytometry

Cell lines to be analyzed with biological replicates were streaked to single colonies and three individual colonies were selected for outgrowth and analysis. Cells were grown in liquid culture as described above, then 500 µL of cell culture was transferred to a microcentrifuge tube and pelleted by centrifugation. Cells were washed once with PBS and then resuspended in 500 µL PBS. Flow cytometry was carried out using either a Guava EasyCyte or EasyCyte HT instrument, collecting >5000 events. Cellular debris and dead cells were excluded on the basis of forward and side scatter, and geometric means of per-cell GFP/RFP fluorescence distributions were used to calculate GFP/RFP for each replicate (*Figures 1–3*) or GFP/RFP ratios were calculated on an individual cell basis for plotting of distributions (Fig *Figure 2—figure supplement 1E-F*). For steady state measurements of GFP-OPT and GFP-CGA reporters in *Figures 1C and 2A* GFP/RFP ratios were further normalized to the mean of WT OPT or WT CGA replicates, respectively, to place them on a similar scale to northern blotting measurements.

## Galactose shutoff RNA half-life assay

Biological replicates of individual cell lines were grown and diluted into 200 mL cultures in SC-URA +2% Gal+2% Raf media as described above. The *not5Δ* strains was typically slow growing and required longer incubations at 30 °C to reach saturation before dilution. When cultures reached $OD_{600}$ 0.4–0.6, they were split into four 50 mL conical tube and pelleted by centrifugation (3000xg, 5 min). Cell pellets were resuspended in 15 mL total prewarmed SC-URA media without added sugar to wash out residual galactose and raffinose and pelleted again by centrifugation in a single 50 mL conical tube. Pellets were resuspended in 10 mL prewarmed SC-URA without added sugar and transferred to a 125 mL beveled flask in a shaking 30 °C incubator. Zero timepoints were taken by removing a 1 mL aliquot of culture, quickly transferring to a microcentrifuge tube and pelleting cells by a snap spin to 4,000xg. Supernatant was decanted and tubes were dropped into liquid nitrogen. To initiate GAL promotor shutoff, 1 mL 40% glucose was added to the 9 mL remaining culture to a final concentration of 4% and a timer was started. Subsequent timepoint samples were taken in a similar manner to the zero timepoint, with the time for each sample recorded at the moment it was dropped into liquid nitrogen. All samples were stored at –80 °C. Downstream RNA extraction and northern blotting proceeded as described below. Plotted replicates represent biological replicates, with the exception of *hel2Δsyh1Δ* minCGA replicate 4, which is a technical replicate of *hel2Δsyh1Δ* minCGA replicate 1.

## Steady-state reporter cell harvesting for northern blot

Cells were grown as described above in 10–15 mL. At $OD_{600}$ 0.4–0.6 cultures were pelleted at 4 °C by centrifugation in a 14 mL culture tube, resuspended in 1 mL PBS (or residual growth media), and transferred to a microcentrifuge tube. Cells were pelleted again by centrifugation at 4 °C and supernatant was decanted. Tubes were dropped into liquid nitrogen and stored at –80 °C until RNA extraction.

## Northern blotting

### RNA extraction

RNA was extracted from frozen cell pellets by hot acid phenol/chloroform extraction. Aliquots of 325 µL acid phenol, pH 4.5 were heated to 65 °C in microcentrifuge tubes on a thermomixer. Cell pellets were retrieved from –80 °C storage and placed on dry ice. Working quickly, individual cell pellets were resuspended in 300–320 µL lysis buffer (8.4 mM EDTA, 60 mM NaOAc pH 5.5, 1.2% SDS) by vortexing just until pellet was fully resuspended. One aliquot of preheated phenol was immediately added to the resuspended pellet and sample was placed onto a thermomixer to minimize time between pellet resuspension and cell lysis. This procedure was repeated for samples being processed in parallel, with each sample shaking at the highest setting on the thermomixer for at least 15 min.

Tubes were then placed in a dry ice-ethanol bath for ~30 s to help precipitate residual SDS and centrifuged at top speed for 3 min. The top aqueous layer was placed in a new tube containing 300 μL room temperature acid phenol. Samples were vortexed several times for a total of 5 min, then centrifuged again at max speed for 30 s. The top aqueous layer was transferred to a tube containing 300 mL room temperature chloroform, vortexed several times for a total of 5 min, and centrifuged at max speed for 3 min. The aqueous phase was then transferred to a tube containing 30 μL 3.5 M NaOAc, pH 5.5. During each step of this process, particular care was taken to avoid transferring any of the organic phase or precipitate at the interface. To each RNA-NaOAc solution, 350 μL of isopropanol was added and mixed well. Tubes were placed on dry ice for at least 30 min, or stored at –80 °C overnight. Samples were spun at top speed in a microcentrifuge for 30 min and the supernatant was aspirated, taking care not to disturb the RNA pellet. Samples were centrifuged again at max speed for 5 min and any remaining supernatant was carefully removed with a 10 μL micropipette. To each RNA pellet, 30 μL of nuclease-free water was added and samples were incubated at 37 °C for 5 min on a thermomixer with gentle shaking to facilitate pellet dissolution. Tubes were then moved to ice and pipetted by hand to ensure full pellet resuspension. Finally, RNA concentrations were measured by a nanodrop spectrophotometer and samples were either used immediately for northern blotting or stored at –80 °C for subsequent use.

## Gel and RNA preparation

A 1.2% agarose formaldehyde gel was prepared by mixing a final concentration of 1 x MOPS electrophoresis buffer, 2.4 g electrophoresis-grade agarose and water to a final volume of 192 mL in a glass 500 mL beaker. This solution was heated in a microwave to boiling and agarose dissolution, mixed, then cooled to approximately 65 °C, placing an insulating material like paper towel beneath to promote even cooling of the solution. Particular care was taken not to allow the agarose to cool further than this before formaldehyde addition, as pieces of unevenly cooled agarose can alter RNA mobility across the gel. When initial cooling was complete, 8 mL 37% formaldehyde and 8 μL ethidium bromide were added and mixed well by swirling. Gel was poured into a mold and allowed to cool fully, then submerged in formaldehyde gel running buffer (1 x MOPS buffer, 1.67% formaldehyde). Meanwhile, RNA samples were prepared by aliquoting an equal mass of total of RNA (typically 10 μg) into microcentrifuge tubes on ice containing an appropriate amount of 5 x RNA loading buffer (bromophenol blue, 4 mM EDTA, 2.66% formaldehyde, 20% glycerol, 30% formamide, 4 x MOPS buffer).

## Gel running and transfer to membrane

RNA samples were boiled at 95 °C for 8 min, then cooled to room temp, spun briefly and loaded onto the gel. Gel was run at 100 V for ~2.5 hr. Gels were imaged on a Typhoon imager to assess RNA quality, then transferred to a Amersham Hybond N+charged nitrocellulose membrane by a BioRad Model 785 Vacuum Blotter following the manufacturer's instructions for transferring RNA, with the alterations of prewetting the membrane with 10 x SSC only and maintaining vacuum between 10–15 inHg. Transfer proceeded for 2 hr. Following transfer, the membrane was carefully removed from the vacuum blotter and placed face up on paper towel for UV crosslinking in a Stratagene UV Stratalinker 2400 on the automatic setting (120 mJ) three times.

## Oligonucleotide probe radiolabeling and hybridization

After crosslinking, the membrane was placed in a glass hybridization bottle with the RNA-side facing away from the glass. Approximately 15 mL Sigma Perfecthyb Plus Hybridization Buffer was added to the bottle and it was placed in a hybridization oven to prewarm for 30 min at 42 °C. Meanwhile, the appropriate oligonucleotide probe was enzymatically radiolabeled with the final reaction concentrations 1 μM oligonucleotide probe, 1 x NEB T4 PNK buffer, 3–6 μL Perkin Elmer gamma-$^{32}$P-ATP, 25 units NEB T4 PNK in a 50 μL reaction volume. This reaction was incubated at 37 °C for 1 hr, then the probe was purified using Cytiva Microspin G-50 columns according to the manufacturer's instructions. The entire volume of probe was then added directly to the prewarmed hybridization solution in the hybridization bottle. Membrane and radiolabeled probe were incubated at 42 °C with rotation overnight. The radioactive hybridization solution was discarded and the membrane was washed three times for 20 min each with ~15 mL low-stringency wash buffer (0.1% SDS, 2 x SSC) at 30 °C. The membrane was placed between transparency film or sheets of plastic wrap and secured into a phosphor storage

screen cassette. A blanked phosphor storage screen was exposed to the radioactive membrane long enough to produce adequate exposure (typically overnight) and imaged as described below. To strip hybridized probe off of the membrane, boiling high-stringency wash buffer (0.1% SDS, 0.2 x SSC) was poured on the membrane in a hybridization bottle, incubated for 10 min at 80 °C, then discarded. The stripping procedure was repeated for a total of two washes, then secondary probing was performed. For experiments with the GFP-OPT, GFP-CGA, and GFP-NONOPT reporters, an oligonucleotide probe for GFP was used as the primary probe and a probe for RFP as the secondary probe. For experiments with minOPT, minCGA, and minNONOPT reporters, a probe for *HIS3* was used as the primary probe and a probe for the endogenous yeast 7 S RNA *SCR1* was used as the secondary probe.

## Phosphor imaging, northern quantification, and half-life calculation

Phosphor storage screens were scanned with a typhoon imager and bands were quantified with ImageQuant TL v8.1 software using rolling ball background subtraction. For galactose shutoff experiments, intensities and timepoints for three or four replicates were fit to a single-exponential decay by least-squares fitting to estimate reporter RNA half-lives. A linear model including strain as a covariate was used to fit the data and perform pairwise comparisons using Tukey's honest significant difference test. For steady-state reporter experiments, either reporter/control ratios (*Figure 1E*, *Figure 2— figure supplement 1C*) were plotted or raw intensity values were fit to a linear mixed model with the design formula with covariates for SCR1 or RFP intensity and knockout strain, and a random covariate for batch (i.e. individual blot). Plotted are adjusted log2(HIS3) values, which are corrected for loading and batch variation and normalized by subtraction to preserve log relationships between samples. The models were tested with pairwise comparisons using Tukey's honest significant difference test.

## Ribosome profiling

### Sample preparation
Ribosome profiling was carried out based on previously published protocols (*McGlincy and Ingolia, 2017*; *Guydosh and Green, 2014*; *Wu et al., 2019b*).

### Culture and ribosome RNA isolation
Cultures were grown to saturation in appropriate media as described above and diluted to $OD_{600}$ 0.1 in 1 L culture. When cells reached $OD_{600}$ 0.4–0.6, cells were harvested by vacuum filtration and pellets were frozen in liquid nitrogen. A portion of each pellet was ground in a SPEX SamplePrep 6870 Freezer/Mill (8 cycles, 10 hz, 1 min run, 1 min cool) with 1 mL pre-frozen lysis buffer (20 mM Tris pH 8, 140 mM KCl, 5 mM $MgCl_2$, 1% Triton X-100, 0.1 mg/mL cycloheximide, 0.1 mg/mL tigecycline) and thawed into 15 mL lysis buffer. Lysates were cleared by centrifugation (5 min, 3000 xg, 4 °C) and supernatants were loaded onto 3 mL sucrose cushion (20 mM Tris pH 8, 150 mM KCl, 5 mM $MgCl_2$, 500 μM DTT, 1 M Sucrose) in a Ti70 ultracentrifuge rotor tube. Samples were centrifuged for 106 min at 60,000 RPM, 4 °C to pellet ribosomes. Supernatant was removed, and the ribosome pellet was rinsed once with lysis buffer excluding cycloheximide and tigecycline (drug-free lysis buffer). Pellet was resuspended by pipetting in 1 mL drug-free lysis buffer. RNA concentrations were measured by Qubit RNA High Sensitivity Assay Kit, 350 μg of RNA was added to a microcentrifuge tube, and volume was increased to at least 400 μL with drug-free lysis buffer. 5 μL Ambion RNaseI was added per 400 μL of RNA solution, and samples were incubated at 25 °C in a thermomixer shaking at 500 RPM too digest free RNA. Samples were placed on ice and 10 μL Superase•In RNase inhibitor was added and mixed to stop the RNase digestion. Sucrose gradients were prepared by a Biocomp Gradient Master (15–40% sucrose gradient containing 20 mM Tris pH 8, 150 mM KCl, 5 mM $MgCl_2$, 500 μM DTT) in SW41 ultracentrifuge rotor tubes and RNase reactions were loaded in top of the gradients. Gradients were centrifuged at 40,000 RPM for 2.5 hr at 4 °C. Gradients were fractionated on a Biocomp Triax gradient fractionator and fractions containing monosomes and disomes were individually pooled and processed in the rest of the downstream protocol. RNA was extracted from samples by SDS-hot phenol/chloroform extraction and isopropanol precipitated with GlycoBlue as co-precipitant.

### Ribosome footprint isolation and reverse transcription
RNA pellets were resuspended in 10 mM Tris pH 7.5 and RNA formamide loading dye and run on a 15% TBE-urea gel, taking care to leave empty lanes between samples to minimize cross-contamination.

Monosome libraries between 15 and 35 nt (monosomes) or 40 and 70 nt (disomes) were cut out of the gel, frozen, eluted overnight in RNA extraction buffer (300 µM NaOAc pH 5.5, 1 mM EDTA pH 8, 0.25% SDS), and precipitated by isopropanol precipitation. RNA pellets were resuspended in 5 µL dephosphorylation reaction mix (7 mM Tris pH 8, 1 x NEB T4 PNK buffer, 10 units Superase•In, 5 units T4 PNK) and incubated at 37 °C for 1 hr. To these reactions, 5 µL of linker ligation mixture was added (38% PEG-8000, 1 x NEB T4 ligase buffer, 2 µM oBZ407_preA preadenylated linker, 100 units NEB T4 RNA ligase 2, truncated) and they were further incubated at 37 °C for 3 hr. Reactions were cleaned up with Zymo Oligo Clean & Concentrator kit and eluted in 10 µL nuclease free water. Samples were supplemented with 1 µL of 10 µM oBZ408 and denatured at 65 °C for 5 min, then placed on ice. To each sample, 8 µL of reverse transcription reaction was added (2.5 x Protoscript II buffer, 12.5 µM DTT, 1.25 mM dNTPs, 20 units Superase•In), samples were mixed, then 1 µL (200 U) Protoscript II reverse transcriptase was added. Samples were incubated 30–60 min at 50 °C, then RNA templates were hydrolyzed by adding 2.2 µL 1 M NaOH and incubating at 95 °C, 5 min. Samples were again purified with Zymo Oligo Clean & Concentrator kit and eluted in 5 µL nuclease free water.

## Ribosomal RNA depletion
A biotinylated subtraction oligo pool from *Guydosh and Green, 2014*, Cell was prepared as in that publication. To each sample, 1 µL of subtraction oligo pool, 1 µL of 20 x SSC, and 2 µL water was added. Oligos were annealed in a thermocycler, denaturing 90 s at 100 °C, then dropping 0.1 °C/s to 37 °C and incubating 15 min. MyOne Streptavidin C1 magnetic beads were prepared for RNA binding per the manufacturers protocol and annealed oligo solutions were transferred to the beads. Solutions were incubated for 15 min at 37 °C, beads were pelleted and supernatants were transferred to new tubes. Samples were cleaned up using Zymo Oligo Clean & Concentrator kit and eluted in 6 µL nuclease free water.

## Final sequencing library preparation
Loading dye was added to samples, and they were run on a 10% TBE-urea gel. With the aid of marker oligos, appropriate sizes were cut out from the gel for each sample and DNA was extracted from gel slices as before with DNA extraction buffer (300 µM NaOAc pH 5.5, 1 mM EDTA pH 8, 10 mM Tris pH8). DNA was isopropanol precipitated, resuspended in 20 µL circularization reaction mix (7.75 mM Tris pH 8, 1 x Epicentre CircLigase buffer, 50 µM ATP, 2.5 mM MnCl$_2$, 50 units CircLigase), incubated at 60 °C for 2 hr and 80 °C for 10 min. Relative cDNA library abundances were tested by qPCR with BioRad iTaq Universal SYBR Green Supermix to identify an appropriate number of PCR amplification cycles for each library. PCR reactions were then performed for the determined number of cycles to introduce sequencing barcodes and amplify libraries (1 x Phusion HF buffer, 200 µM dNTPs, 0.5 µM oBZ287 universal forward PCR primer, 1 µM reverse barcode PCR primer, 7.5% v/v cDNA template, 1 unit Phusion polymerase). Samples were mixed with loading dye and loaded on an 8% TBE PAGE gel. Gel was run 60 min, 200 V and each library was cut from the gel, frozen, and extracted from the gel slice as above using DNA gel extraction buffer. Libraries were isopropanol precipitated, resuspended in 6 µL 10 mM Tris pH 8, and assessed for quality and concentration using an Agilent BioAnalyzer 2100 High Sensitivity DNA assay. Libraries were pooled and sequenced at the Johns Hopkins University Genetic Resources Core Facility on an Illumina NovaSeq 6000 instrument.

## Data processing
Reads from raw FASTQ files were trimmed and aligned using a custom Python script to run software from the BBtools suite (https://jgi.doe.gov/data-and-tools/bbtools/) and the STAR aligner (*Dobin et al., 2013*). Subsequent analyses were performed by custom Python scripts. Briefly, reads on start codons from all genes in monosome libraries were used to calculate distances from the 5' end of a read to the ribosomal A site (17 nt for monosomes, 50 nt for disomes). Reads per million mapped reads (RPMs) were calculated at each position of the reporter RNAs by dividing the number of A site shifted 5' ends at a given position by the total number of reads mapped to the genome (not including those that mapped to ncRNA). Reads with lengths 19–26 nt were considered part of the 21-mer population and reads with lengths 27–35 nt were considered part of the 28-mer population. Ratios of 21-mers/28-mers were calculated by dividing read numbers of 21-mers on the reporter (excluding the FLAG tag, five codons upstream of the top codon and the common binding region of the northern

blotting probe) by read numbers of 28-mers on the reporter and normalizing to the 21-mer/28-mer ratio for all genes in a sample (*Figure 5E*). Ratios of 21-mers/28-mers in the CGA region of the minCGA reporter were calculated by re-aligning unaligned reads allowing multimapping (STAR option --outFilterMultimapNmax 999), then excluding any reads outside the CGA region and counting each unique read only once.

### Gene set enrichment analysis (GSEA)

Using data from the GFP-CGA screen ranked by per-plate Z-score without LOESS normalization, ranked GSEA was performed using the GSEApy library for Python, querying the GO Biological Process annotation (*GSEApy, 2022*; *Xie et al., 2021*; *Subramanian et al., 2005*). The top five most enriched terms in each direction were selected for plotting. Full results are available in the supplemental files.

### Affinity purification-mass spectrometry (AP-MS)

TAP-tagged Syh1 and Smy2 strains were purchased from Dharmacon and grown as described above. TAP tag purifications were performed as previously published (*Amberg et al., 2006*) replacing NP-40 for Triton X-100 and excluding TCA precipitation. Samples were submitted to the Johns Hopkins University Mass Spectrometry and Proteomics Core facility and processed by facility personnel as follows. Samples were reduced with DTT, alkylated with iodoacetamide and FASP digested on a 30 kDa filter with 10 ng/µl trypsin in 25 mM TEAB at 37 ° C overnight. Peptides were step-fractionated by basic reverse phase chromatography on a µ-HLB Oasis plate. Samples were loaded in 0.1% TFA, eluted with 10 mM TEAB containing 5, 15, 20, 25, or 50% acetonitrile and fractions were dried. Each fraction was reconstituted in 2% acetonitrile and 0.1% formic acid and injected for MS/MS.

Raw data produced by the core facility was analyzed by MaxQuant (*Tyanova et al., 2016*) searching against the UniProt yeast database and LFQ values for identified proteins were calculated without imputation, combining data from all fractions of each sample.

### Multiple sequence alignment

Structure-aware multiple sequence alignment for human GIGYF1 (NCBI accession: O75420.2), GIGYF2 (NCBI accession: Q6Y7W6.1), and yeast Syh1 (NCBI accession: NP_015220.1) and Smy2 (NCBI accession: NP_015220.1) was performed by T-Coffee Expresso (*Notredame et al., 2000*). An additional alignment was performed by EMBL-EBI MUSCLE (https://www.ebi.ac.uk/Tools/msa/muscle) to independently verify alignment results. T-Coffee Expresso alignment results were processed with ESPript 3.0 (*Robert and Gouet, 2014*) and output was included as *Figure 2—figure supplement 1G*. Regions corresponding to the DDX6 binding motif identified in *Weber et al., 2020* were shaded.

### Data availability

Ribo-seq data is available in the NCBI Gene Expression Omnibus (GEO) (https://www.ncbi.nlm.nih.gov/geo/) database with the accession GSE189404. The mass spectrometry proteomics data have been deposited to the ProteomeXchange Consortium via the PRIDE (*Perez-Riverol et al., 2017*) partner repository with the dataset identifier PXD030076. Code to process sequencing data is available at https://github.com/greenlabjhmi/2022_syh1/, (copy archived at swh:1:rev:-2f26e680e06d11fd517105d52f19005f00b5c7af; *Veltri, 2022*).

## Acknowledgements

We thank Allen R Buskirk, Niladri Sinha, and Nicolle Rosa Mercado for careful reading of the manuscript and all Green lab members for helpful discussions throughout this study. High throughput sequencing was performed at the Johns Hopkins Genetic Resources Core Facility (RRID: SCR_018669) and the Johns Hopkins Single Cell and Transcriptomics Core. Mass spectrometry was performed by the Johns Hopkins Mass Spectrometry Core. Funding Canadian Institutes of Health Research Foundation Grant FDN-159913 Grant W Brown, National Institutes of Health (R37GM059425) Rachel Green, National Institutes of Health (5T32GM135131-02) Juliette Lecomte.

## Additional information

### Competing interests

Karole N D'Orazio: is affiliated with Regeneron Pharmaceuticals. Laura N Lessen: is affiliated with GlaxoSmithKline. The other authors declare that no competing interests exist.

### Funding

| Funder | Grant reference number | Author |
| --- | --- | --- |
| Canadian Institutes of Health Research | FDN-159913 | Grant W Brown |
| National Institutes of Health | R37GM059425 | Rachel Green |
| National Institutes of Health | 5T32GM135131-02 | Anthony J Veltri |

The funders had no role in study design, data collection and interpretation, or the decision to submit the work for publication.

### Author contributions

Anthony J Veltri, Conceptualization, Data curation, Software, Formal analysis, Investigation, Visualization, Methodology, Writing – original draft, Project administration, Writing – review and editing; Karole N D'Orazio, Conceptualization, Data curation, Formal analysis, Investigation, Writing – review and editing; Laura N Lessen, Data curation, Formal analysis, Investigation, Methodology, Writing – review and editing; Raphael Loll-Krippleber, Supervision, Investigation, Methodology, Project administration, Writing – review and editing; Grant W Brown, Resources, Supervision, Investigation, Methodology, Writing – review and editing; Rachel Green, Conceptualization, Resources, Supervision, Funding acquisition, Methodology, Writing – original draft, Writing – review and editing

### Author ORCIDs

Anthony J Veltri ⓘ http://orcid.org/0000-0002-7067-1796
Grant W Brown ⓘ http://orcid.org/0000-0002-9002-5003
Rachel Green ⓘ http://orcid.org/0000-0001-9337-2003

### Decision letter and Author response

Decision letter https://doi.org/10.7554/eLife.76038.sa1
Author response https://doi.org/10.7554/eLife.76038.sa2

---

## Additional files

### Supplementary files

• Supplementary file 1. A Microsoft Excel table containing raw data from the R-SGA GFP-reporter screens, including an additional sheet describing overlaps of screen hits.

• Supplementary file 2. A CSV table containing GFP/RFP fluorescence ratios for validation flow cytometry from GFP-CGA screen hits.

• Supplementary file 3. A CSV table of statistically significant validated GFP-CGA screen hits including p-values and Z-scores.

• Supplementary file 4. A CSV table describing the gene-set enrichment analysis (GSEA) of the GFP-CGA screen resulting from analysis with the software gseapy.

• Supplementary file 5. A CSV table with p-values for pairwise statistical comparisons for all relevant figures using linear modeling and Tukey's Honest Significant Difference test.

• Supplementary file 6. A CSV table containing LFQ values for mass spectrometry analysis.

• Supplementary file 7. A CSV table containing average mRNA half-lives and standard deviations for minOPT, minCGA, and minNONOPT reporters in various knockout strains.

• Supplementary file 8. A CSV table of oligos used in the study.

• Supplementary file 9. A CSV table of plasmids used in the study.

- Supplementary file 10. A CSV table of yeast strains used in the study.
- Transparent reporting form

## Data availability

Ribo-seq data is available in the NCBI Gene Expression Omnibus (GEO) (https://www.ncbi.nlm.nih.gov/geo/) database with the accession GSE189404. The mass spectrometry proteomics data have been deposited to the ProteomeXchange Consortium via the PRIDE (Perez-Riverol et al. 2019) partner repository with the dataset identifier PXD030076. Code to process sequencing data is available at https://github.com/greenlabjhmi/2022_syh1/, (copy archived at swh:1:rev:2f26e680e06d11fd517105d52f19005f00b5c7af).

The following datasets were generated:

| Author(s) | Year | Dataset title | Dataset URL | Database and Identifier |
|---|---|---|---|---|
| Veltri AJ, Green R | 2021 | Distinct ribosome states trigger diverse mRNA quality control pathways | https://www.ncbi.nlm.nih.gov/geo/query/acc.cgi?&acc=GSE189404 | NCBI Gene Expression Omnibus, GSE189404 |
| Veltri AJ, Green R | 2021 | Distinct ribosome states trigger diverse mRNA quality control pathways | https://www.ebi.ac.uk/pride/archive?keyword=PXD030076 | PRIDE, PXD030076 |

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
