## [Editor Report]

This study provides a broad comparison of the roles of protein factors in No-Go Decay (NGD) and Codon-Optimality-Mediated Decay (COMD) in the yeast, *S. cerevisiae*. A major strength of the manuscript is the direct comparison between one mRNA with a single strong translational stall and another similar mRNA with many slow translation sites (caused by changes in the genetic code). The analysis of both the factors that cause decay of these mRNAs as well as the ribosome states on the different mRNAs increases our understanding of the molecular basis for the different mechanisms of mRNA quality control. The results support a model that these are largely distinct processes driven by different protein factors in response to different ribosome conformations and, in addition, the work identifies several differences between yeast and human NGD and COMD.

---

## [Decision Letter]

**Decision letter after peer review:**

Thank you for submitting your article "Distinct ribosome states trigger diverse mRNA quality control pathways" for consideration by *eLife*. Your article has been reviewed by 3 peer reviewers, including Alan Hinnebusch as the Reviewing Editor and Reviewer #1, and the evaluation has been overseen by James Manley as the Senior Editor. The reviewers have opted to remain anonymous.

Essential revisions:

There was agreement among the referees and reviewing editor that an important strength of the paper involves (i) the genetic data comparing the effects of different mutations on the NGD and COMD reporters, indicating that NGD and COMD involve largely distinct factors, with the exception of Not5 that functions in both pathways; and (ii) the ribosome profiling data on these reporters revealing that the appearance of collided ribosomes at a discrete stall site applies only to the NGD reporter.

1) However, there was also agreement that these profiling data are not adequate to conclude that the different ribosomal states are necessarily the triggers for NGD or COMD, but merely provide correlative data consistent with this possibility and, as such, the title of the paper and last sentence of the Abstract should be modified to more accurately reflect what these data actually show.

2) There are important concerns about the second major conclusion, that Syh1 is the key effector of the NGD pathway in wild-type cells, and the claim that new mechanistic insights have been provided into Syh1 function. The genetic data are quite complex, with some ostensibly contradictory results involving Hel2; and the data seem to point instead to a model that Syh1 functions in only one of two pathways with extensively overlapping functions in NGD, the other involving Hel2 and Cue2, with the Syh1 pathway being only somewhat more important than the Cue2 pathway, such that mRNA decay is affected when Syh1 is deleted from otherwise WT cells. The authors should consider including this alternative interpretation of the results.

3) There are additional genetic experiments that are likely needed to support aspects of the final model in Figure 6, including that Xrn1 functions only in the Syh1 pathway, and that the Cue2 pathway requires Hel2, as described in Ref. #1's comments. These requests for additional work are viewed as being justified, not only to provide stronger support for the final model, but also because some of the key observations have been published previously by this group or other labs, including the involvement of Syh1 itself in NGD. As such, the genetic analysis of syh1 mutations should be as thorough as possible.

4) A related issue is whether Smy2 might have an important overlapping function with Syh1, which should be addressed by determining whether there is a strong NGD phenotype on deleting Smy2 in a double mutant lacking Syh1 (see Ref. #3 comments).

5) The referees also took issue with the claim that new mechanistic insight into Syh1 function has been provided, as it remains unclear whether Syh1 is actually recruited by collided ribosomes. This question could be addressed biochemically, or perhaps genetically by combining syh1 mutations (actually a hel2 syh1 mutation) with mutations in GCN1 or MBF1. If the authors elect not to carry out these experiments, they should soften their claims about providing a mechanistic understanding of Syh1 function and indicate in the Discussion that additional work is needed to determine whether Syh1 is actually recruited to collided ribosomes in the manner depicted in Figure 6.

6) Finally, it was felt that the results of the reporter screen of the yeast deletion collection conducted using the COMD reporter should be presented with the list of mutations that tested positive in that screen, and to compare it to the list of mutations similarly identified using the NGD reporter, which would increase the scientific value of the study.

*Reviewer #1 (Recommendations for the authors):*

– The authors wish to conclude that Syh1 represents the major pathway for NGD in WT cells, and that Cue2 functions only when the ability of Slh1 to diminish collided ribosomes is overwhelmed. I wasn't convinced of this interpretation as it seems equally possible that the two pathways are continuously functioning in a largely redundant manner, but with Syh1 making a moderate unique contribution even when Cue2 is present, while being able to fully compensate for loss of the Cue2 pathway in cells lacking either Cue2 or Hel2. On lines 350-352, they argue that "in the syh1∆ strain, NGD reporter levels are somewhat rescued because recruitment of Xrn1 is impaired, but Cue2-mediated endonucleolytic decay plays a larger role as Slh1 becomes overwhelmed". Why would Slh1 become overwhelmed in the absence of Syh1? In fact, deleting SLH1 reduces minCGA expression in cells lacking SYH1, so it doesn't seem to be overwhelmed.

– There were some additional mutants that should be analyzed to support their final model in Figure 6. The syh1 xrn1 double mutant should be examined to show that the minCGA reporter will be derepressed identically as in each of the syh1 or xrn1 single mutants, as predicted by the conclusion that Syh1 is essential for turnover of the CGA reporter by Xrn1. (In fact, it appears that the GFP-OPT reporter is derepressed to a greater extent by deletion of XRN1 vs deletion of SYH1, at odds with the model.) In addition, the derepression observed in the xrn1 single mutant should be less than observed in the syh1 cue2 double mutant, as the latter should be defective for both the Syh1/Xrn1 and Hel2/Cue2 pathways.

– Is there any independent evidence that Cue2 function requires Hel2? If not, then a hel2 cue2 syh1 triple is required, which should phenocopy the hel2 syh1 and cue2 syh1 double mutants in the manner predicted if Hel2 and Cue2 function, as proposed, in the same pathway.

– If it was not shown previously for the minCGA reporter, it is important to verify that deleting CUE2 will lead to a large derepression of this reporter in cells deleted of SLH1.

– The data in Figure 2 show that deleting SLH1 reduces minCGA mRNA levels in the absence of SYH1. Does this mean that increased ribosome collisions in cells lacking Slh1 also enhance turnover through the Cue2/Hel2 pathway? The authors should address this finding.

– Results from quantifying the 3' frag of the minCGA reporter seem to suggest different conclusions compared to those derived from data in Figure 2. Can the authors explain why?

– It seems important to show that the deletion of NOT5 or XRN1 will increase expression of the NONOPT GFP reporter as a positive control for the Figure 3B results.

– Figure 1D and Figure 2C: the authors should perform tests and indicate whether mean values differ statistically in cases where it is not obvious that the means differ between mutant and WT or between two different mutants, where such differences would be important for their final model.

– The authors' explanation of the impact of hel2Δ mutation on collided ribosomes on NGD reporters is hard to follow. On the one hand, hel2Δ seems to elevate NGD by impairing ubiquitylation of the collided ribosomes, thereby increasing collisions and enhancing NGD. On the other hand, hel2Δ is predicted to impair NGD mediated by the Cue2 endonuclease. They also conclude that WT Hel2 stabilizes disomes but facilitates their removal at collision sites-if they are stabilized by Hel2 why do they get removed? Can the authors attempt to resolve these apparent discrepancies in the Discussion?

*Reviewer #2 (Recommendations for the authors):*

Suggestions for improved or additional experiments, data or analyses, especially those directed at increasing the impact of the work and making it suitable for *eLife*.

The experimental and analytical work presented is very good and generally supports the authors conclusions. I am not completely convinced that these conclusions will have an impact suitable for *eLife*, but I think it is arguable that they could.

1. It seems that the SGA screens did not identify any new factors in NGD or COMD, though this is not completely clear in the manuscript. Can you please specifically state whether or not the study identified new factors involved in these pathways?

2. The positive hits for the COMD screen (nonopt) are not described at all. This is a major limitation. There are many points in the Figure 3C volcano plot that look like strong hits. A comprehensive report on the SGA results from nonopt vs opt reporters could help increase the impact.

3. Moreover, Venn diagrams of shared hits from the NGD and COMD SGA experiments would help the reader discern the extent to which the pathways are distinct.

*Reviewer #3 (Recommendations for the authors):*

The manuscript by Veltri et al. describes the use of a genome-wide screen of non-essential yeast genes to identify genes that modulate expression of a reporter with a strong translational stall [GFP-HIS3-(CGA)12]. These studies provide additional evidence that Syh1 and Smy2 proteins (homologs of mammalian GIGYF1/2 proteins) as well as the ribosomal protein Asc1 modulate mRNA decay of a reporter with a strong translational stall site; deletion of these genes results in increased expression of the reporter. Surprisingly in this screen, deletions of canonical No Go Decay (NGD) factors HEL2, SLH1, CUE3 and RQT4 result in reduced expression of the same reporter (the opposite of their expected effects). The validity of SYH1, HEL2 and SLH1 mutant effects are demonstrated by reconstruction of the deletions and demonstration that both protein and mRNA are affected as expected. To begin to understand the interaction of Syh1 with the NGD pathway and possibly the basis for its recruitment to the stalling reporter, the authors make double mutants between SYH1 and the NGD components. They find that Syh1 function in mRNA decay does not require function of the major NGD regulator Hel2, of the NGD endonuclease Cue2 or of the ribosome disassociation factor Slh1. In fact, double mutants in which SYH1 is deleted in conjunction with either HEL2 or CUE2 result in substantially increased mRNA from a HIS3-CGA stalling reporter compared to the single SYH1 deletion (similar to mRNA levels of HIS3 reporter without the stall site). By contrast, none of these factors affect mRNA stability of a HIS3 reporter in which translation elongation is likely uniformly slowed by suboptimal codons (HIS3-NONOPT). However, as expected, HIS3-NONOPT is a target of COMD (codon optimality mediated decay) and is stabilized by deletion of NOT5. In fact, the strong stalling reporter, HIS3-CGA, is also stabilized in the NOT5 deletion, in a manner quantitatively similar to its stabilization by the SYH1 deletion. Thus, mRNA stability of the strong stalling reporter is regulated by both NGD and COMD. To understand the molecular basis for recruitment of distinct decay factors, the authors investigate the ribosome states on these reporters using ribosome profiling. In both the OPT and NONOPT reporters, single ribosomes and disomes (collided ribosomes) are uniformly positioned across the coding sequence, while in the CGA reporter, ribosomes are absent from the region downstream of the CGA repeats and disomes are substantially increased and stacked at the CGA repeat (compared to the OPT reporter). In addition, ribosomes lacking tRNA from the A site are enriched on the NONOPT reporter and with the CGA codon repeat on the CGA reporter. The authors infer that the distinct ribosome signals of collided disomes or ribosomes with empty A sites are strong determinants of the factors that lead to mRNA decay.

The paper does not meet the claim in the abstract and body of the paper that Syh1 acts as the primary link to mRNA decay in NGD nor does it provide new mechanistic insight into role of Syh1 in NGD (line 23).

1. It is unclear if Syh1 primarily acts in the NGD pathway, basically it is unclear if the (CGA)12 reporters are primarily targeted by NGD since the deletions of the known NGD factors either have no effect or cause decreased stability of the (CGA)12 reporters. For instance, see line 155 "This focused analysis of protein and RNA reporter levels in these different strains validates results from the initial screen and indicates that the CGA reporter is being strongly regulated by canonical NGD machinery." However, the results with Hel2 and Slh1 are not the expected results of deleting factors with a key role in NGD (and moreover the HEL2 effects are not recapitulated with the HIS3 minireporter). The authors should address this discrepancy, at least by citing the evidence that this reporter is an NGD substrate for Cue2 (in D'Orazio et al).

2. The genetic epistasis further implies that Syh1 does not act as the primary nuclease in the NGD pathway. Figure 2- effects of syh1∆ on mRNA from HIS3-CGA12 reporter is less than a 2-fold effect, but double deletions syh1∆ hel2∆ and syh1∆ cue2∆ result in >7-fold increase in mRNA (close to reporter lacking CGA codons). Why conclude that Syh1 is the primary NGD factor? Genetics here indicates two redundant pathways-if one inactivates both, then the mRNA is stable. Primarily this result is correctly discussed in the Discussion, but is not really proven (see 3)

3. The choice to focus on only SYH1 (because its effects were stronger than SMY2) is questionable. It is possible to miss many scenarios in which one protein substitutes for the other or to miss the full spectrum of their importance. Since the most novel result in this manuscript is the involvement of Syh1 in NGD, it seems incumbent to test the homologous gene which also affects NGD. At least examine the effects of a double reconstructed mutant syh1∆ smy2∆ double deletions (with and without deletion of hel2) and show the effects of the single reconstructed smy2∆.

4. It is unclear what type of mechanistic insight is provided since the manuscript does not supply any information on the mechanism by which Syh1 is recruited to the target mRNA nor its molecular mode of action. The test of the two hypotheses about the recruitment are not complete. The deletion of MBF1 might activate the Cue2-dependent pathway, resulting in apparent low levels of GFP/RFP protein. However, there is no simple way to prove this point. If this were true, a double deletion of mbf1 and hel2 would completely stabilize the mRNA. This experiment or the one below are valuable because they provide some additional evidence that Syh1 is recruited by collided ribosomes.

5. The authors do not consider the possibility that Syh1 is recruited to collided ribosomes by Gcn1 or other components that bind collided ribosomes with Gcn1 (Gcn20, Rbg2, Gir2). Syh1 recruitment could be mediated by either the Gcn1 complex or Hel2 or both. Given the papers from the Guydosh and Zaher labs documenting the competition between the NGD and GAAC pathways (as well as the Gcn1 complex on collided ribosome Pochopien and Wilson paper), it seems incumbent upon the authors to test this possibility. This reviewer did note that GCN1, GCN20… deletions did not result in increased expression (high throughput data). It is possible that Syh1 can be recruited by either Gcn1 or Hel2 (or that the mutants in the deletion collection have acquired suppressors- a known problem with the collection.) Either of these points, might also be addressed with biochemical evidence that Syh1 recognizes collided ribosomes.

6. The paper provides insufficient evidence to meet the claim in the title and body to "define the molecular triggers that determine how distinct pathways target mRNAs for degradation in yeast (line 24)." There are multiple differences in the ribosome profiling of the different reporters, but no evidence of a causal relationship. One might consider simply revising the title and softening the claim here. The claim that ribosome collisions are key to Syh1 recruitment could be tested either by reducing ribosome collisions on a CGA reporter with different 5' UTRs (see Simms and Zaher 2019) or by reducing global initiation. The effects of the SYH1 deletion on these reporters could be measured as well as ribosome profiling- it is unclear if one would also need to reduce the number of CGA codons. (The authors might also look at the mass spec enrichment of two subunits of eIF3 in the pull down of Syh1).

[Editors' note: further revisions were suggested prior to acceptance, as described below.]

Thank you for resubmitting your work entitled "Distinct elongation stalls during translation trigger distinct pathways for mRNA degradation" for further consideration by *eLife*. Your revised article has been evaluated by James Manley (Senior Editor) and a Reviewing Editor.

The revised manuscript is much improved; however, there are a few remaining issues.

First, Figure 2 and its description still have some difficulties:

– Lines 203-205 and lines 398-399: these statements are not justified as the difference between WT and syh1 is not statistically significant (as shown in File S5). In fact, the syh1 mutation only has a strong effect on the minCGA reporter in cells lacking HEL2 or CUE2, underscoring the extensive functional redundancy between the Syh1 and Hel2/Cue pathways, at least for this reporter, which should be more fully acknowledged at these two locations as is done later on in the DISCUSSION. The Abstract would also be more accurate/informative if it stated explicitly the functional redundancy between the Syh1 and Hel2/Cue2 pathways.

– Lines 205-210 and lines 401-402: this description is not accurate as neither the syh1 nor smy2 mutations significantly rescued the mRNA level vs. WT, whereas the double mutation does. These findings don't justify a focus on Syh1 vs Smy2, nor exclusion of Smy2 from their model for NGD in WT cells. The greater importance of Syh1 vs. Smy2 emerges only in the next paragraph when the hel2 syh1 and hel2 smy2 double mutants are compared. Moreover, because the difference in mRNA level between hel2smy2 vs hel2 is significant, it should be acknowledged that Smy2 and Syh1 have overlapping functions with Syh1 playing merely the greater role. Including the latter point in the Abstract is also recommended.

– Lines 244-247: these data seem to be overinterpreted as there are no statistically significant differences in mRNA levels between the syh1mbf1 strain and either the syh1 or mbf1 single mutant strains. Thus, it would be safer to conclude that there is no evidence from these assays that Mbf1 is involved in either the Hel2/Cue2 or Syh1/Smy2 pathways.

– Overall, the Abstract and Discussion would be more accurate/informative if they conveyed the functional overlaps between Syh1 and Smy2, as well as the redundancy between the Syh1/Smy2 and Hel2/Cue2 pathways in NGD. The revised Discussion does do a good job of explaining this issue regarding Syh1 vs Hel2/Cue2, but it would be even better to acknowledge a significant, even if minor, contribution of Smy2 to the Syh1 pathway.

Second, it would be better to modify the title of the paper to replace the word "trigger" with "are associated (or linked) with", to fall in line with the wording changes made in the Abstract that softened this conclusion in the revised version of the paper.

---

## [Author Response]

Essential revisions:There was agreement among the referees and reviewing editor that an important strength of the paper involves (i) the genetic data comparing the effects of different mutations on the NGD and COMD reporters, indicating that NGD and COMD involve largely distinct factors, with the exception of Not5 that functions in both pathways; and (ii) the ribosome profiling data on these reporters revealing that the appearance of collided ribosomes at a discrete stall site applies only to the NGD reporter.1) However, there was also agreement that these profiling data are not adequate to conclude that the different ribosomal states are necessarily the triggers for NGD or COMD, but merely provide correlative data consistent with this possibility and, as such, the title of the paper and last sentence of the Abstract should be modified to more accurately reflect what these data actually show.

We agree that the ribosome profiling data do not directly demonstrate a connection between ribosome state and NGD / COMD and have altered the manuscript to reflect the correlative nature of our conclusions regarding ribosome states triggering mRNA decay.

2) There are important concerns about the second major conclusion, that Syh1 is the key effector of the NGD pathway in wild-type cells, and the claim that new mechanistic insights have been provided into Syh1 function. The genetic data are quite complex, with some ostensibly contradictory results involving Hel2; and the data seem to point instead to a model that Syh1 functions in only one of two pathways with extensively overlapping functions in NGD, the other involving Hel2 and Cue2, with the Syh1 pathway being only somewhat more important than the Cue2 pathway, such that mRNA decay is affected when Syh1 is deleted from otherwise WT cells. The authors should consider including this alternative interpretation of the results.

We agree that the claim that Syh1 is the major pathway in NGD was too strong. Our revised manuscript argues that the data can be explained by the presence of parallel NGD pathways dependent on either Syh1 or Hel2. We have altered the language in the results and Discussion sections to reflect this possibility and describe the parallel activities of Slh1/Cue2 in compensating for loss of Syh1.

3) There are additional genetic experiments that are likely needed to support aspects of the final model in Figure 6, including that Xrn1 functions only in the Syh1 pathway, and that the Cue2 pathway requires Hel2, as described in Ref. #1's comments. These requests for additional work are viewed as being justified, not only to provide stronger support for the final model, but also because some of the key observations have been published previously by this group or other labs, including the involvement of Syh1 itself in NGD. As such, the genetic analysis of syh1 mutations should be as thorough as possible.

The dependence of Cue2 function on Hel2 was previously established in D’Orazio et al. 2019 and Ikeuchi et al. 2019a and nicely reviewed in D’Orazio et al. 2021a. These studies show that Cue2-dependent endonuclease cleavages are lost when Hel2 is deleted. Furthermore, D’Orazio et al. 2019 established that Cue2 activity is increased in genetic backgrounds where Slh1 is deleted; we interpreted this to mean that Slh1 (known to function as a helicase) normally ‘rescues’ collided ribosomes, but that Cue2 acts as a failsafe mechanism that cleaves mRNAs when stalls are not resolved by Slh1. Of direct relevance to this manuscript, we suggest that the decrease of minCGA mRNA levels observed in the *SLH1* deletion background is the consequence of increased cleavage by Cue2. Indeed, when *CUE2* is knocked out in the *slh1∆syh1* background (i.e. the triple mutant *cue2∆slh1∆syh1∆*) there is a strong rescue of reporter levels, highlighting the role of CUE2 in cleaving these mRNAs (Figure 2C). We have updated the manuscript to more explicitly describe the (previously established) dependence of Cue2 on Hel2 and to clarify our genetic argument based on the previous literature.

We agree with the reviewer that double knockouts of *XRN1* and NGD factor genes are important experiments to establish the direct connection between Syh1- and Xrn1-mediated decay. In an effort to address these points, we constructed additional strains with the minOPT and minCGA reporters including *xrn1∆, syh1∆xrn1∆*, and *hel2∆xrn1∆.* While we were able to build these strains, we were concerned by their extremely impaired growth rate and thus their potential to pick up suppressor mutations over the course of genetic manipulation and reporter insertion. Indeed, we observed inconsistent control reporter levels between biological replicates in these strains and therefore are not able to make conclusions based on these experiments. We have altered the language throughout the manuscript to soften our claims about a direct connection between Syh1 activity and Xrn1-mediated RNA decay.

4) A related issue is whether Smy2 might have an important overlapping function with Syh1, which should be addressed by determining whether there is a strong NGD phenotype on deleting Smy2 in a double mutant lacking Syh1 (see Ref. #3 comments).

We agree that further justification of our choice to focus on SYH1 was warranted. We performed the suggested experiments including SMY2 knockouts and double mutants (now Figure 2—figure supplement 1B, shown here). The results of this experiment indicate that, while SMY2 knockout does increase minCGA reporter levels in a manner consistent with our previously published data (Hickey, et al. 2020), the strong rescue of minCGA reporter levels observed in the *hel2∆syh1∆* strain is not observed in a *hel2∆smy2∆* strain and the *hel2∆syh1∆smy2∆* strain shows only a small increase compared to the *hel2∆smy2∆* strain. Thus, while Smy2 does appear to contribute to decay in a similar fashion to Syh1, its contribution appears to be relatively minor, thus justifying our focus on Syh1.

5) The referees also took issue with the claim that new mechanistic insight into Syh1 function has been provided, as it remains unclear whether Syh1 is actually recruited by collided ribosomes. This question could be addressed biochemically, or perhaps genetically by combining syh1 mutations (actually a hel2 syh1 mutation) with mutations in GCN1 or MBF1. If the authors elect not to carry out these experiments, they should soften their claims about providing a mechanistic understanding of Syh1 function and indicate in the Discussion that additional work is needed to determine whether Syh1 is actually recruited to collided ribosomes in the manner depicted in Figure 6.

We agree with the reviewers. In order to more fully address this point, we performed an additional experiment (see Author response image 1) to assay recruitment of tagged Syh1 protein to collided ribosomes. We used an optimized concentration of the translation elongation inhibitor cycloheximide to induce global ribosome collisions. Then we performed sucrose gradient fractionation and immunoblotting to look for the recruitment of tagged Syh1 protein to the ribosome collisions. In this assay, we did not observe a substantial deep shift in the distribution of Syh1 in the samples with induced ribosome collisions. This may indicate that Syh1 is only transiently recruited to ribosome collisions or that it is not directly recruited to ribosome collisions. Further experiments are needed to determine the nature of potential physical interactions between the ribosome and Syh1 in the context of NGD. We have updated the discussion to indicate that further work is needed on this front.

To test the effects of Mbf1 on minCGA reporter levels in combination with Syh1, we performed additional genetic experiments, including the construction of *MBF1* double knockouts with *HEL2* and *SYH1* (Figure S2E, shown below). If Syh1 is recruited to ribosome collisions by Mbf1 in yeast in a similar manner to the recruitment of GIGYF2 by EDF1 in mammalian cells, we expect that *MBF1* deletion strains will behave similarly to *SYH1* deletion strains, across genetic backgrounds. While the *mbf1∆* strain does increase minCGA reporter mRNA levels to a similar modest extent as *syh1∆*, we note that the *hel2∆mbf1∆* strain does not reveal the same strong increase in reporter levels as the *hel2∆syh1∆* strain. In fact, *hel2∆mbf1∆* minCGA mRNA levels are similar to those of *mbf1∆* alone. Additionally, reporter levels are moderately increased in the *syh1∆mbf1∆* strain compared to *mbf1∆* or *syh1∆* alone, suggesting an additive effect of these knockouts and potential parallel pathways (similar to the genetic interaction between *SYH1* and *HEL2*). These results raise the interesting possibility that Mbf1 may actually function as part of the Hel2-dependent NGD pathway rather than in Syh1-dependent NGD. We have included these new data and a more careful discussion surrounding these ideas.

**Author response image 1. sa2fig1:** . Polysome association of SYH1-TAP protein. Yeast cells containing c-terminally TAP-tagged SYH1 at the endogenous locus were grown to saturation in YPD, diluted to OD600 0.1 and allowed to reach OD600 0.5. Cells were then treated with either 2.8 µg/mL cycloheximide or DMSO in YPD media for 30 minutes. Yeast were collected by vacuum filtration, lysed via freezer mill in polysome lysis buffer. Lysates were thawed, cleared by centrifugation, and either RNase treated and run on 10-35% sucrose gradients to verify increase in disomes (A) or directly fractionated on 10-50% sucrose gradients (B). Protein from undigested fractions was TCA precipitated, resuspended and run on an SDS-PAGE gel. Proteins were transferred to PVDF membrane and immunoblotted using Anti-TEV antibody (Thermo PA1-119) and an HRP conjugated secondary antibody (C).

6) Finally, it was felt that the results of the reporter screen of the yeast deletion collection conducted using the COMD reporter should be presented with the list of mutations that tested positive in that screen, and to compare it to the list of mutations similarly identified using the NGD reporter, which would increase the scientific value of the study.

We have included further description of the top screen hits in the text and have included an additional sheet in supplementary table 1 describing the screen hits and the overlap.

Reviewer #1 (Recommendations for the authors):– The authors wish to conclude that Syh1 represents the major pathway for NGD in WT cells, and that Cue2 functions only when the ability of Slh1 to diminish collided ribosomes is overwhelmed. I wasn't convinced of this interpretation as it seems equally possible that the two pathways are continuously functioning in a largely redundant manner, but with Syh1 making a moderate unique contribution even when Cue2 is present, while being able to fully compensate for loss of the Cue2 pathway in cells lacking either Cue2 or Hel2. On lines 350-352, they argue that "in the syh1∆ strain, NGD reporter levels are somewhat rescued because recruitment of Xrn1 is impaired, but Cue2-mediated endonucleolytic decay plays a larger role as Slh1 becomes overwhelmed". Why would Slh1 become overwhelmed in the absence of Syh1? In fact, deleting SLH1 reduces minCGA expression in cells lacking SYH1, so it doesn't seem to be overwhelmed.

See full response to editor above.

– There were some additional mutants that should be analyzed to support their final model in Figure 6. The syh1 xrn1 double mutant should be examined to show that the minCGA reporter will be derepressed identically as in each of the syh1 or xrn1 single mutants, as predicted by the conclusion that Syh1 is essential for turnover of the CGA reporter by Xrn1. (In fact, it appears that the GFP-OPT reporter is derepressed to a greater extent by deletion of XRN1 vs deletion of SYH1, at odds with the model.) In addition, the derepression observed in the xrn1 single mutant should be less than observed in the syh1 cue2 double mutant, as the latter should be defective for both the Syh1/Xrn1 and Hel2/Cue2 pathways.

See full response to editor above.

– Is there any independent evidence that Cue2 function requires Hel2? If not, then a hel2 cue2 syh1 triple is required, which should phenocopy the hel2 syh1 and cue2 syh1 double mutants in the manner predicted if Hel2 and Cue2 function, as proposed, in the same pathway.– If it was not shown previously for the minCGA reporter, it is important to verify that deleting CUE2 will lead to a large derepression of this reporter in cells deleted of SLH1.– The data in Figure 2 show that deleting SLH1 reduces minCGA mRNA levels in the absence of SYH1. Does this mean that increased ribosome collisions in cells lacking Slh1 also enhance turnover through the Cue2/Hel2 pathway? The authors should address this finding.

As discussed above, previous studies established that Cue2 is fully dependent on Hel2 for function and that in the absence of Slh1 that the role of Cue2 increases. These studies used similar reporters containing CGA12 repeats to demonstrate these effects (D’Orazio et al. 2019 and Ikeuchi et al. 2019a).

– Results from quantifying the 3' frag of the minCGA reporter seem to suggest different conclusions compared to those derived from data in Figure 2. Can the authors explain why?

We believe the model from D’Orazio et al. 2019 explains both the full length and 3’ fragment reporter data for the minCGA reporter. The 3’ reporter fragment can result from two different sources: (1) 5’ to 3’ degradation by Xrn1 that is blocked by stalled ribosomes near the stall site and (2) cleavage by Cue2 between collided ribosomes and slow clearing of ribosomes downstream of cleavage. Cue2 activity depends on Hel2 and normally contributes little to decay of the reporter, except in backgrounds where Slh1 (and thus ribosome clearing) is impaired. This explains why there is little accumulation of the 3’ fragment, except in backgrounds where Slh1 is deleted (*slh1∆, slh1∆syh1∆*) and thus where Cue2 plays a dominant role -- the bands in this case are a combination of Cue2 cleavage fragments and Xrn1-produced fragments. The triple knockout *cue2∆slh1∆syh1∆* strain has reduced 3’ fragment accumulation compared to the *slh1∆syh1∆* strain due to the loss of cleavage by Cue2, but the modest contribution of Xrn1 to 3’ fragment production is still present, as stalled ribosomes are not cleared. We have added additional clarification to these points in the manuscript to highlight the consistency of these results with the full length minCGA reporter results.

– It seems important to show that the deletion of NOT5 or XRN1 will increase expression of the NONOPT GFP reporter as a positive control for the Figure 3B results.

While we agree that this would be a clarifying detail to report, the reality is more complicated. Despite the stabilization of the GFP-NONOPT reporters that we (and others) show in various genetic backgrounds (NOT4, XRN1 and DHH1), the mRNA levels of these reporters are not increased in these same backgrounds. This has long been interpreted to reflect some sort of compensation by the cell, likely transcriptional, to maintain mRNA levels in an independent manner (Bregman, et al., *Cell*, 2011, DOI: 10.1016/j.cell.2011.12.005). This complexity was beyond the scope of this study, but likely explains the lack of interesting candidates to emerge in the GFP screen based on steady state expression levels of the GFP-NONOPT reporter. See Author response image 2.

**Author response image 2. sa2fig2:** (A-C) Flow cytometry analysis of GFP-containing reporters in the indicated genetic backgrounds. In all cases, there was little change between the GFP/RFP levels for WT and knockout GFP-NONOPT reporters.

– Figure 1D and Figure 2C: the authors should perform tests and indicate whether mean values differ statistically in cases where it is not obvious that the means differ between mutant and WT or between two different mutants, where such differences would be important for their final model.

We have included statistical tests for key data where indicated in the figures, and included full pairwise statistics in a supplemental table.

– The authors' explanation of the impact of hel2Δ mutation on collided ribosomes on NGD reporters is hard to follow. On the one hand, hel2Δ seems to elevate NGD by impairing ubiquitylation of the collided ribosomes, thereby increasing collisions and enhancing NGD. On the other hand, hel2Δ is predicted to impair NGD mediated by the Cue2 endonuclease. They also conclude that WT Hel2 stabilizes disomes but facilitates their removal at collision sites-if they are stabilized by Hel2 why do they get removed? Can the authors attempt to resolve these apparent discrepancies in the Discussion?

Based on the literature and our data, we suggest that the role of Hel2 is to stabilize the collided ribosome interface which is important for recruitment of downstream factors Slh1 and Cue2, explaining both the importance of Hel2 for promoting NGD and our interpretation that it stabilizes the disome structure. We suggest that in the *HEL2* deletion strain there are decreased levels of the CGA reporter (in our R-SGA experiment and in the follow up reporter experiments) because of the compensatory effects of the Syh1-mediated NGD pathway. Additional language clarifying these views has been added to the discussion.

Reviewer #2 (Recommendations for the authors):Suggestions for improved or additional experiments, data or analyses, especially those directed at increasing the impact of the work and making it suitable for eLife.The experimental and analytical work presented is very good and generally supports the authors conclusions. I am not completely convinced that these conclusions will have an impact suitable for eLife, but I think it is arguable that they could.1. It seems that the SGA screens did not identify any new factors in NGD or COMD, though this is not completely clear in the manuscript. Can you please specifically state whether or not the study identified new factors involved in these pathways?2. The positive hits for the COMD screen (nonopt) are not described at all. This is a major limitation. There are many points in the Figure 3C volcano plot that look like strong hits. A comprehensive report on the SGA results from nonopt vs opt reporters could help increase the impact.3. Moreover, Venn diagrams of shared hits from the NGD and COMD SGA experiments would help the reader discern the extent to which the pathways are distinct.

We have clarified in the Results section the high-confidence hits that emerged from our NGD screen and follow up validation. In particular, 14 of our top hits are now explicitly named in the text with a brief description of their function, and a full list of the 18 high-confidence hits from our validation are presented in an additional supplemental table. As the reviewer points out, many of these hits have previously been implicated in NGD, while the GIGY proteins are novel hits from this screen.

The COMD screen identified no strong hits that were validated, likely because even factors that might stabilize these mRNAs (such as Dhh1 and Not5) do not generally increase their steady state levels (discussed above). Nevertheless, we have added further analysis of the COMD screen, including GSEA analysis and presentation of the screen results in a supplementary table. We have also included additional Venn diagrams such as Figure 3—figure supplement 1 to help compare the CGA, COMD, and AAA reporter screens. There is very limited overlap between the top genes from the COMD screen with either the CGA or AAA screen hits.

Reviewer #3 (Recommendations for the authors):The manuscript by Veltri et al. describes the use of a genome-wide screen of non-essential yeast genes to identify genes that modulate expression of a reporter with a strong translational stall [GFP-HIS3-(CGA)12]. These studies provide additional evidence that Syh1 and Smy2 proteins (homologs of mammalian GIGYF1/2 proteins) as well as the ribosomal protein Asc1 modulate mRNA decay of a reporter with a strong translational stall site; deletion of these genes results in increased expression of the reporter. Surprisingly in this screen, deletions of canonical No Go Decay (NGD) factors HEL2, SLH1, CUE3 and RQT4 result in reduced expression of the same reporter (the opposite of their expected effects). The validity of SYH1, HEL2 and SLH1 mutant effects are demonstrated by reconstruction of the deletions and demonstration that both protein and mRNA are affected as expected. To begin to understand the interaction of Syh1 with the NGD pathway and possibly the basis for its recruitment to the stalling reporter, the authors make double mutants between SYH1 and the NGD components. They find that Syh1 function in mRNA decay does not require function of the major NGD regulator Hel2, of the NGD endonuclease Cue2 or of the ribosome disassociation factor Slh1. In fact, double mutants in which SYH1 is deleted in conjunction with either HEL2 or CUE2 result in substantially increased mRNA from a HIS3-CGA stalling reporter compared to the single SYH1 deletion (similar to mRNA levels of HIS3 reporter without the stall site). By contrast, none of these factors affect mRNA stability of a HIS3 reporter in which translation elongation is likely uniformly slowed by suboptimal codons (HIS3-NONOPT). However, as expected, HIS3-NONOPT is a target of COMD (codon optimality mediated decay) and is stabilized by deletion of NOT5. In fact, the strong stalling reporter, HIS3-CGA, is also stabilized in the NOT5 deletion, in a manner quantitatively similar to its stabilization by the SYH1 deletion. Thus, mRNA stability of the strong stalling reporter is regulated by both NGD and COMD. To understand the molecular basis for recruitment of distinct decay factors, the authors investigate the ribosome states on these reporters using ribosome profiling. In both the OPT and NONOPT reporters, single ribosomes and disomes (collided ribosomes) are uniformly positioned across the coding sequence, while in the CGA reporter, ribosomes are absent from the region downstream of the CGA repeats and disomes are substantially increased and stacked at the CGA repeat (compared to the OPT reporter). In addition, ribosomes lacking tRNA from the A site are enriched on the NONOPT reporter and with the CGA codon repeat on the CGA reporter. The authors infer that the distinct ribosome signals of collided disomes or ribosomes with empty A sites are strong determinants of the factors that lead to mRNA decay.The paper does not meet the claim in the abstract and body of the paper that Syh1 acts as the primary link to mRNA decay in NGD nor does it provide new mechanistic insight into role of Syh1 in NGD (line 23).1. It is unclear if Syh1 primarily acts in the NGD pathway, basically it is unclear if the (CGA)12 reporters are primarily targeted by NGD since the deletions of the known NGD factors either have no effect or cause decreased stability of the (CGA)12 reporters. For instance, see line 155 "This focused analysis of protein and RNA reporter levels in these different strains validates results from the initial screen and indicates that the CGA reporter is being strongly regulated by canonical NGD machinery." However, the results with Hel2 and Slh1 are not the expected results of deleting factors with a key role in NGD (and moreover the HEL2 effects are not recapitulated with the HIS3 minireporter),. The authors should address this discrepancy, at least by citing the evidence that this reporter is an NGD substrate for Cue2 (in D'Orazio et al).

We have included additional references to evidence from D’Orazio et al. 2019 supporting the targeting of our CGA-repeat-containing reporters by NGD. Further, we have clarified in the text our interpretation that loss of *HEL2* and *SLH1* decrease CGA reporter levels because of the presence of multiple parallel decay pathways (i.e. SYH1-mediated decay and CUE2-mediated decay). We have also clarified our interpretation of the apparent discrepancy between the CGA and minCGA reporters in the *hel2∆* strain. While loss of *HEL2* has a reduced effect on the minCGA reporter, we believe this may be the result of different stalling contributions of the P2A-containing CGA reporter (the P2A sequence itself may cause ribosome stalling to some extent). Importantly, we do observe a decrease of minCGA reporter levels in the *slh1∆* strain, consistent with the CGA reporter and our previous work demonstrating increased activity of the endonuclease Cue2 on NGD reporters in an *slh1∆* strain (Figure 2C). Consistent with this interpretation, the levels of the 3’ fragment of the minCGA reporter (resulting from Cue2 cleavage and 5’ to 3’ decay) are greatly increased in the *slh1∆* strain (Figure 2—figure supplement 1A).

2. The genetic epistasis further implies that Syh1 does not act as the primary nuclease in the NGD pathway. Figure 2- effects of syh1∆ on mRNA from HIS3-CGA12 reporter is less than a 2-fold effect, but double deletions syh1∆ hel2∆ and syh1∆ cue2∆ result in >7-fold increase in mRNA (close to reporter lacking CGA codons). Why conclude that Syh1 is the primary NGD factor? Genetics here indicates two redundant pathways-if one inactivates both, then the mRNA is stable. Primarily this result is correctly discussed in the Discussion, but is not really proven (see 3)

We agree with the reviewer’s argument and have altered our conclusions to reflect the redundant and compensatory nature of the Syh1 and Hel2/Cue2 mediated NGD pathways.

3. The choice to focus on only SYH1 (because its effects were stronger than SMY2) is questionable. It is possible to miss many scenarios in which one protein substitutes for the other or to miss the full spectrum of their importance. Since the most novel result in this manuscript is the involvement of Syh1 in NGD, it seems incumbent to test the homologous gene which also affects NGD. At least examine the effects of a double reconstructed mutant syh1∆ smy2∆ double deletions (with and without deletion of hel2) and show the effects of the single reconstructed smy2∆.

We performed an additional set of experiments with *SMY2* knockouts to support our choice in focusing on Syh1. See full response to editor above.

4. It is unclear what type of mechanistic insight is provided since the manuscript does not supply any information on the mechanism by which Syh1 is recruited to the target mRNA nor its molecular mode of action. The test of the two hypotheses about the recruitment are not complete. The deletion of MBF1 might activate the Cue2-dependent pathway, resulting in apparent low levels of GFP/RFP protein. However, there is no simple way to prove this point. If this were true, a double deletion of mbf1 and hel2 would completely stabilize the mRNA. This experiment or the one below are valuable because they provide some additional evidence that Syh1 is recruited by collided ribosomes.

See full response to editor above.

5. The authors do not consider the possibility that Syh1 is recruited to collided ribosomes by Gcn1 or other components that bind collided ribosomes with Gcn1 (Gcn20, Rbg2, Gir2). Syh1 recruitment could be mediated by either the Gcn1 complex or Hel2 or both. Given the papers from the Guydosh and Zaher labs documenting the competition between the NGD and GAAC pathways(as well as the Gcn1 complex on collided ribosome Pochopien and Wilson paper), it seems incumbent upon the authors to test this possibility. This reviewer did note that GCN1, GCN20… deletions did not result in increased expression (high throughput data). It is possible that Syh1 can be recruited by either Gcn1 or Hel2 (or that the mutants in the deletion collection have acquired suppressors- a known problem with the collection.) Either of these points, might also be addressed with biochemical evidence that Syh1 recognizes collided ribosomes.

See full response to editor above.

6. The paper provides insufficient evidence to meet the claim in the title and body to "define the molecular triggers that determine how distinct pathways target mRNAs for degradation in yeast (line 24)." There are multiple differences in the ribosome profiling of the different reporters, but no evidence of a causal relationship. One might consider simply revising the title and softening the claim here. The claim that ribosome collisions are key to Syh1 recruitment could be tested either by reducing ribosome collisions on a CGA reporter with different 5' UTRs (see Simms and Zaher 2019) or by reducing global initiation. The effects of the SYH1 deletion on these reporters could be measured as well as ribosome profiling- it is unclear if one would also need to reduce the number of CGA codons. (The authors might also look at the mass spec enrichment of two subunits of eIF3 in the pull down of Syh1).

See full response to editor above.

[Editors' note: further revisions were suggested prior to acceptance, as described below.]

The revised manuscript is much improved; however, there are a few remaining issues.First, Figure 2 and its description still have some difficulties:– Lines 203-205 and lines 398-399: these statements are not justified as the difference between WT and syh1 is not statistically significant (as shown in File S5). In fact, the syh1 mutation only has a strong effect on the minCGA reporter in cells lacking HEL2 or CUE2, underscoring the extensive functional redundancy between the Syh1 and Hel2/Cue pathways, at least for this reporter, which should be more fully acknowledged at these two locations as is done later on in the DISCUSSION. The Abstract would also be more accurate/informative if it stated explicitly the functional redundancy between the Syh1 and Hel2/Cue2 pathways.

We have updated the language in the indicated sections to be in agreement with the statistical tests performed on the reporter levels. We have also included language in the abstract highlighting the redundancy between the Syh1/Smy2 pathway and the Hel2-mediated pathway.

– Lines 205-210 and lines 401-402: this description is not accurate as neither the syh1 nor smy2 mutations significantly rescued the mRNA level vs. WT, whereas the double mutation does. These findings don't justify a focus on Syh1 vs Smy2, nor exclusion of Smy2 from their model for NGD in WT cells. The greater importance of Syh1 vs. Smy2 emerges only in the next paragraph when the hel2 syh1 and hel2 smy2 double mutants are compared. Moreover, because the difference in mRNA level between hel2smy2 vs hel2 is significant, it should be acknowledged that Smy2 and Syh1 have overlapping functions with Syh1 playing merely the greater role. Including the latter point in the Abstract is also recommended.

We have made changes in these locations and in the abstract and discussion to more clearly recognize the contributions of Smy2 to NGD as indicated by our assays.

– Lines 244-247: these data seem to be overinterpreted as there are no statistically significant differences in mRNA levels between the syh1mbf1 strain and either the syh1 or mbf1 single mutant strains. Thus, it would be safer to conclude that there is no evidence from these assays that Mbf1 is involved in either the Hel2/Cue2 or Syh1/Smy2 pathways.

The conclusion that Mbf1 may be involved in the Hel2-mediated pathway has been removed to be in accordance with the statistical significance of the assays.

– Overall, the Abstract and Discussion would be more accurate/informative if they conveyed the functional overlaps between Syh1 and Smy2, as well as the redundancy between the Syh1/Smy2 and Hel2/Cue2 pathways in NGD. The revised Discussion does do a good job of explaining this issue regarding Syh1 vs Hel2/Cue2, but it would be even better to acknowledge a significant, even if minor, contribution of Smy2 to the Syh1 pathway.

As noted above, additional language has been added to the abstract to highlight the functional overlap between Syh1 and Smy2, as well as to emphasize the redundancy of the Syh1/Smy2-mediated and Hel2/Cue2-mediated pathways in NGD.

Second, it would be better to modify the title of the paper to replace the word "trigger" with "are associated (or linked) with", to fall in line with the wording changes made in the Abstract that softened this conclusion in the revised version of the paper.

The title has been updated to more correctly reflect the conclusions of our study.